# Acute inhibition of neurosteroid estrogen synthesis suppresses status epilepticus in an animal model

Satoru M Sato, Catherine S Woolley*

Department of Neurobiology, Northwestern University, Evanston, United States

**Abstract** Status epilepticus (SE) is a common neurological emergency for which new treatments are needed. In vitro studies suggest a novel approach to controlling seizures in SE: acute inhibition of estrogen synthesis in the brain. Here, we show in rats that systemic administration of an aromatase (estrogen synthase) inhibitor after seizure onset strongly suppresses both electrographic and behavioral seizures induced by kainic acid (KA). We found that KA-induced SE stimulates synthesis of estradiol (E2) in the hippocampus, a brain region commonly involved in seizures and where E2 is known to acutely promote neural activity. Hippocampal E2 levels were higher in rats experiencing more severe seizures. Consistent with a seizure-promoting effect of hippocampal estrogen synthesis, intra-hippocampal aromatase inhibition also suppressed seizures. These results reveal neurosteroid estrogen synthesis as a previously unknown factor in the escalation of seizures and suggest that acute administration of aromatase inhibitors may be an effective treatment for SE.

## Introduction

Status epilepticus (SE) is a common neurological emergency characterized by a prolonged seizure episode. ~One-third of SE cases are refractory to first line (benzodiazepine) and second line (anti-epileptic drug) treatments, and often require pharmacologically induced coma to control seizures (*Novy et al., 2010*; *Betjemann and Lowenstein, 2015*). The overall mortality rate of SE is estimated at 20% (*DeLorenzo et al., 1995*) and patients who recover have an increased likelihood of subsequent unprovoked seizures (*Hesdorffer et al., 1998*; *Holtkamp et al., 2005*). Thus there is a need for new approaches to acute seizure control.

Estrogens have long been known to be proconvulsant, both in animal models (*Terasawa and Timiras, 1968*; *Nicoletti et al., 1985*; *Buterbaugh and Hudson, 1991*; *Edwards et al., 1999*; *Woolley, 2000*) and in human patients (*Bäckström, 1976*; *Herzog et al., 1997*). Most studies have assumed that the estrogens relevant to seizures are of ovarian origin and so have studied estrogen regulation of seizures on the time scale of the reproductive cycle. However, accumulating evidence suggests that synthesis of estrogens in the brain as neurosteroids could acutely promote neural activity, especially in the hippocampus, a limbic brain region commonly involved in seizures.

The hippocampus of both sexes contains enzymatic machinery to synthesize estrogens, including 17β-estradiol (E2) (*Hojo et al., 2004*; *Kretz et al., 2004*; *Tabatadze et al., 2014*), and studies in primary hippocampal cultures (*Prange-Kiel et al., 2003*), slice cultures (*Kretz et al., 2004*) and acute slices (*Hojo et al., 2004*) show that the hippocampus can synthesize E2 in vitro. In hippocampal slices, E2 levels were increased by 30 min of NMDA treatment (*Hojo et al., 2004*), suggesting that neurosteroid E2 synthesis may be stimulated by neural activity. In vitro studies also show that E2 acutely promotes neural activity. For example, E2 applied to hippocampal slices increases neuronal excitability (*Teyler et al., 1980*; *Kumar and Foster, 2002*; *Carrer et al., 2003*; *Wu et al., 2011*),

*For correspondence: cwoolley@ northwestern.edu

**Competing interests:** The authors declare that no competing interests exist.

**eLife digest** Seizures occur when connected groups of cells in the brain become over-active and fire together. Current anti-seizure medications work by reducing brain activity generally. Although this is often effective in controlling seizures, it can also lead to negative side effects like drowsiness, dizziness or difficulty concentrating. A better alternative would be to target a factor that promotes activity especially during seizures.

Most people think of estrogens as being female sex hormones. However, estrogens are also made in the brain of both sexes, where they could promote activity during seizures. Sato and Woolley therefore set out to test a two-part hypothesis: that seizures stimulate the production of estrogen in the brain, and that inhibiting this production process just as seizures begin would make seizures less severe.

Sato and Woolley studied male and female rats and found that in both sexes, seizures stimulate the production of estrogens in the hippocampus – a part of the brain that is often involved in seizures. Because estrogens are known to increase the activity of cells in the hippocampus, this suggested that estrogens that are produced in the brain during seizures could make seizures worse. Sato and Woolley tested this by injecting rats with a drug that inhibits estrogen production, called an aromatase inhibitor, shortly after seizures began. The drug strongly suppressed seizures, whereas control rats that did not receive the injection continued to have seizures.

Overall, Sato and Woolley show that the production of estrogen in the brain escalates seizure activity, and suggest that aromatase inhibitors may be useful for controlling seizures. Several questions remain that require further study. How does seizure activity lead to estrogen being made in the brain? How do estrogen levels go back down after a seizure? What circumstances other than seizures stimulate brain estrogen production, and what roles does this production process play in activity that is not related to seizures?

potentiates excitatory synaptic transmission (*Wong and Moss, 1992*; *Kramar et al., 2009*; *Smejkalova and Woolley, 2010*; *Oberlander and Woolley, 2016*), and suppresses inhibitory synaptic transmission (*Huang and Woolley, 2012*; *Tabatadze et al., 2015*), on a time scale of minutes. Thus, in vitro studies indicate both that neural activity could stimulate neurosteroid E2 synthesis and that E2 acutely promotes neural activity.

Based on these findings, we hypothesized that, in vivo, the onset of seizures initiates a positive feedback loop in which neural activity stimulates neurosteroid estrogen synthesis, and that in turn, synthesized estrogens potentiate neural activity further, contributing to seizure escalation during SE. To investigate this, we used a combination of electrographic and behavioral seizure testing, aromatase (estrogen synthase) inhibitor treatment, and microdialysis in male and female rats injected with kainic acid (KA), a well-established animal model of SE (*Lévesque and Avoli, 2013*). The results demonstrated that neurosteroid estrogen synthesis is seizure-promoting, in both sexes, and that acutely inhibiting aromatase, either systemically or specifically within the hippocampus, strongly suppresses seizures. Thus neurosteroid estrogen synthesis is a previously unknown factor that contributes to seizures in SE, and acute treatment with an aromatase inhibitor may be an effective new approach to seizure control.

## Results

### Extra-gonadal aromatase inhibition acutely suppresses seizures

We first sought to determine whether acutely inhibiting extra-gonadal synthesis of estrogens affects seizures in one or both sexes. To eliminate gonadal steroids, animals were gonadectomized. Male (n=9) and female (n=12) rats were randomly assigned to receive vehicle or the selective aromatase inhibitor, fadrozole (*Häusler et al., 1989a*; *1989b*; *Browne et al., 1991*), and were injected with KA (10 mg/kg, i.p.; *Sperk et al., 1985*) to induce SE. After seizures were detected in hippocampal EEG recordings (*Figure 1—figure supplement 1A–D*), vehicle or fadrozole (40 mg/kg) was administered through a tail vein catheter, and seizures were recorded for an additional 2 hrs (*Figure 1A*). The

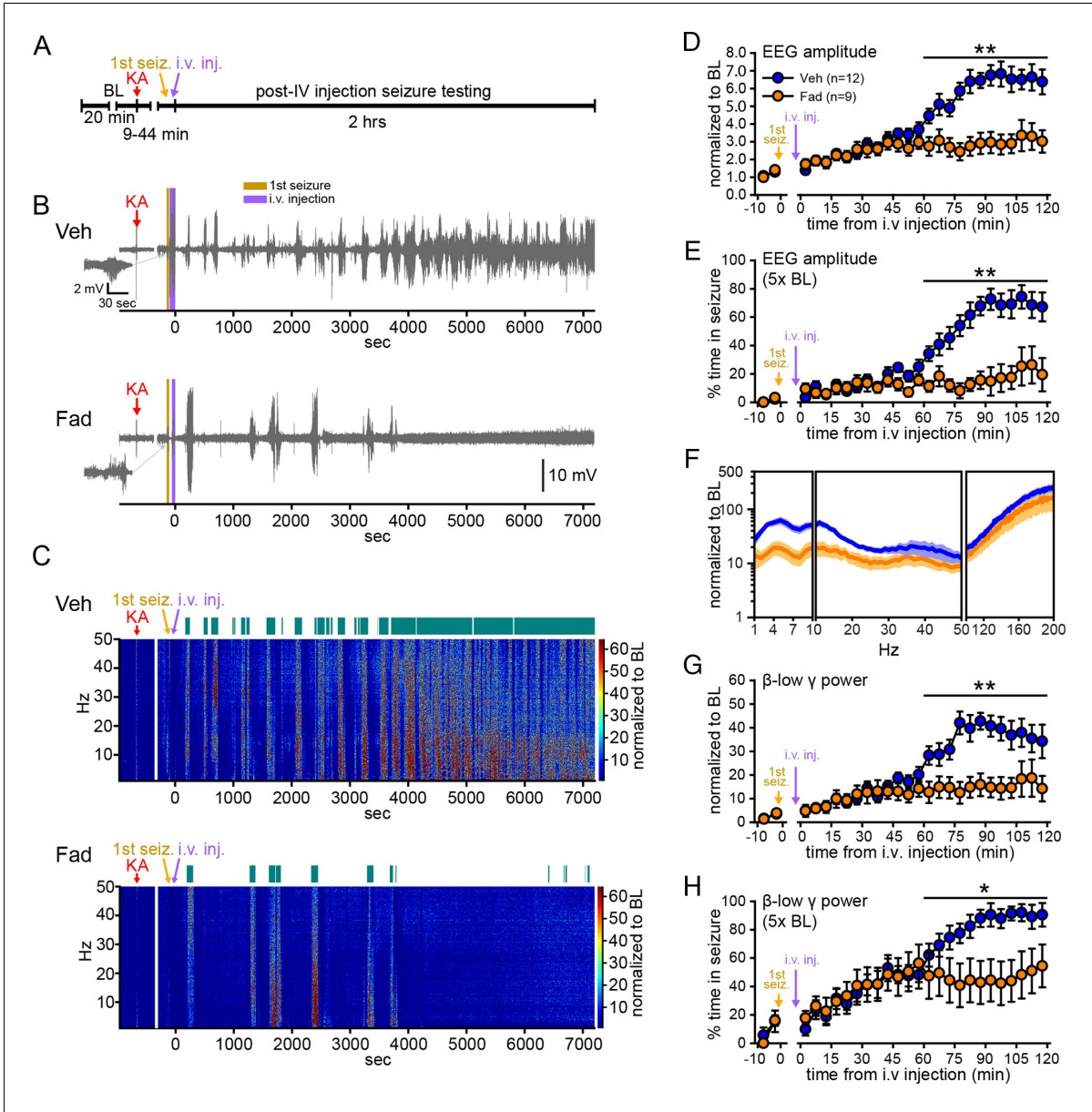

**Figure 1.** Systemic inhibition of aromatase after seizure onset suppresses electrographic seizures in gonadectomized rats. (A) Schematic of experimental design. (B, C) Representative EEG (B) and normalized power spectrum (C) from one vehicle (Veh, top)- and one fadrozole (Fad, bottom)-treated rat. For EEG, the 1st seizure in each animal is shown on the left. Teal bars above heatmaps indicate seizures detected by 5x baseline power in the β-low γ (10–50 Hz) range as the threshold. (D) Mean ± SEM normalized EEG amplitude plotted in 5 min bins for vehicle (blue, n=12)- and fadrozole (orange, n=9)-treated rats. The escalation of seizures evident in vehicle-treated rats during the 2nd hr of testing was inhibited by fadrozole. (E) Mean ± SEM percent time in seizure using 5x baseline thresholds of EEG amplitude showing that the progressive increase in time in seizure evident in vehicle-treated rats was also inhibited by fadrozole. (F) Mean ± SEM normalized power spectrogram plotted in 1 Hz bins for δ-θ (1–10 Hz), β-low γ (10–50 Hz), and ripple (100–200 Hz) frequency ranges for vehicle (blue) and fadrozole (orange) rats showing the KA-induced increase in power relative to baseline for each frequency range examined. Increases in δ-θ (p<0.001, ANOVA) and β-low γ (p=0.01, ANOVA) power were lower in fadrozole- than vehicle-treated rats. (G) Mean ± SEM normalized EEG β-low γ power plotted in 5 min bins for vehicle (blue) and fadrozole (orange) rats showing the lack of seizure escalation in fadrozole-treated rats. (H) Mean ± SEM percent time in seizure plotted in 5 min bins using 5x baseline thresholds of β-low γ power for vehicle (blue) and fadrozole (orange) rats also showing the lack of seizure escalation in fadrozole-treated rats. *p<0.05 and **p<0.01 between vehicle- and fadrozole-treated rats, post-hoc unpaired t-tests. BL = baseline.

The following figure supplements are available for figure 1:

**Figure supplement 1.** No difference in characteristics of the 1st seizure in animals subsequently treated with vehicle or fadrozole.

*Figure 1 continued on next page*

*Figure 1 continued*

**Figure supplement 2.** Comparison of manual and 3x, 5x, and 10x baseline thresholds.

dose of fadrozole was chosen based on previous studies (*Wade et al., 1994*; *Taziaux et al., 2007*) and results of a pilot study (see Materials and methods). Seizures detected prior to i.v. injections did not differ in latency, duration, amplitude, or β-low γ power in rats assigned to either treatment group (*Figure 1—figure supplement 1E–I*).

*Figure 1B and C* show representative recordings of electrographic seizures throughout the 2 hr seizure monitoring period. Vehicle-treated rats showed the expected progression of seizure activity, beginning with individual seizures that evolved to merging seizures (*Treiman et al., 1990*) with high EEG amplitude and increased power. In contrast, fadrozole-treated rats showed much lower seizure activity, particularly during the $2^{nd}$ hr when vehicle rats typically experienced merging seizures. Thus fadrozole suppressed seizure escalation. The effects of fadrozole were the same in males and females (all p values >0.30) so data were combined. The seizure-induced increase in EEG amplitude during the $2^{nd}$ hr was 46% lower on average in fadrozole-treated compared to vehicle-treated rats (time x drug: $F_{1,19}=10.36$, p<0.01, *Figure 1D*) and the time animals spent in seizures based on a 5x baseline threshold of EEG amplitude was 73% lower in the $2^{nd}$ hr (time x drug: $F_{1,19}=20.88$, p<0.001, *Figure 1E*). In addition to EEG amplitude, fadrozole also significantly attenuated the seizure-induced increase in EEG power in the δ-θ (1–10 Hz, $F_{1,19}=21.50$, p<0.001) and β-low γ (10–50 Hz, $F_{1,19}=8.08$, p=0.01) frequency ranges over the entire 2 hr testing period (*Figure 1F*). Additional analyses revealed that the effect on power was more prominent during the $2^{nd}$ hr of seizure testing, when normalized power was significantly lower in all frequency ranges examined (time x drug: δ-θ: $F_{1,19}=20.86$, p<0.001; β-low γ: $F_{1,19}=20.22$, p<0.001; ripple (100–200 Hz): $F_{1,19}=5.96$, p<0.05, not shown). We focused further analyses on β-low γ power because this is known to be sensitive to KA both in vitro (*Fisahn et al., 2004*) and in vivo (*Medvedev et al., 2000*; *Lévesque et al., 2009*) and may contribute to the spread of seizures to other brain regions (*Finnerty and Jefferys, 2000*; *Lévesque et al., 2009*). The seizure-induced increase in β-low γ power in fadrozole-treated rats was 59% lower during the $2^{nd}$ hr (*Figure 1G*) and time in seizure based on a threshold of 5x baseline β-low γ power was 44% lower in the $2^{nd}$ hr (time x drug: $F_{1,19}=16.87$, p<0.01, *Figure 1H*). The same results for time in seizure were obtained using 3x or 10x baseline thresholds of EEG amplitude or β-low γ power (*Figure 1—figure supplement 2*). These results demonstrated that inhibition of extra-gonadal aromatase suppressed electrographic seizure activity in the hippocampus.

Fadrozole also suppressed behavioral seizures. Behavioral seizures from the same vehicle- and fadrozole-treated rats as in *Figure 1B* are shown in *Figure 2A*. KA seizures are characterized by stereotyped and progressively more severe seizure behaviors commonly identified using a Racine scale (*Racine, 1972*). These seizures typically progress from periods of immobility and staring (Racine 0), to chewing (Racine 1), head-waving (Racine 2), forelimb clonus (Racine 3), rearing (Racine 4), and falling/tonic-clonic seizure (Racine 5), although the severity of seizures varies among individual animals. The seizure behaviors that appear earliest (Racine 0–2) correspond with electrographic seizure activity in the hippocampus (*Figure 2B,C*) and reflect mild limbic seizures, in contrast to the later-occurring convulsive seizures (Racine 3–5), which reflect both limbic and extra-limbic seizure activity (*Lothman and Collins, 1981*; *McIntyre et al., 1982*; *Lothman et al., 1989*; *Handforth and Ackermann, 1995*).

Because electrographic and behavioral seizures were recorded in the same animals, we were able to evaluate whether fadrozole affected the relationship between specific seizure behaviors and hippocampal EEG activity. Cross-correlation between EEG and the occurrence of specific seizure behaviors for each rat confirmed that normalized hippocampal EEG amplitude ($F_{4,86}=30.22$, p<0.001; *Figure 2D*) and power ($F_{4,86}=15.63$, both p<0.001; *Figure 2E*) are better correlated with behavioral signs of mild limbic seizures (Racine 0, 1–2) than convulsive seizures (Racine 3, 4, 5). Interestingly, fadrozole decreased the strength of cross-correlation between EEG and some behaviors, particularly those reflecting convulsive seizures (Racine 3: amplitude: $t_{18}=2.87$, power: $t_{18}=2.80$, and Racine 4: power: $t_{17}=2.57$, all p<0.05). Thus systemic fadrozole not only suppressed seizure activity, but also altered the temporal relationship between hippocampal EEG activity and convulsive seizures.

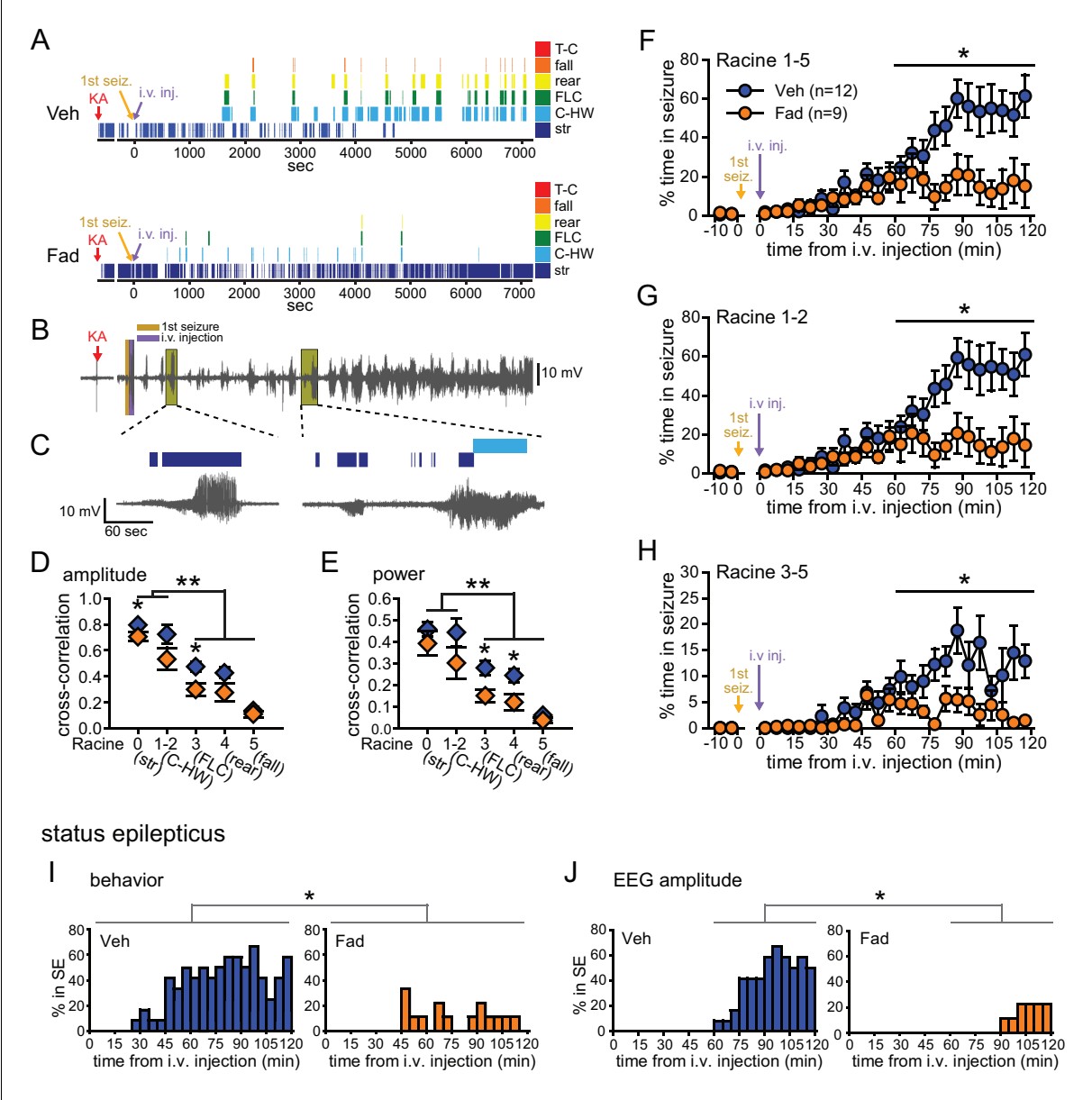

**Figure 2.** Systemic inhibition of aromatase after seizure onset suppresses behavioral seizures and disrupts EEG-behavior relationships in gonadectomized rats. (A) Representative behavioral seizures from the same vehicle (top)- and fadrozole (bottom)-treated rats as in *Figure 1B* plotted per sec. Each seizure behavior is indicated with a different color: staring (str, Racine 0, dark blue), chewing and head waving (C-HW, Racine 1–2, light blue), forelimb clonus (FLC, Racine 3 green), rearing (rear, Racine 4, yellow), falling (fall, Racine 5, orange), and tonic-clonic seizure (T-C, Racine 5, red). (B) EEG recording from the vehicle-treated rat shown in (A). (C) One early and one middle electrographic seizure are magnified to show their correspondence with mild limbic seizure behaviors using the same behavioral markers as in (A). (D, E) Cross-correlation of normalized hippocampal EEG amplitude (D) and β-low γ power (E) was stronger for mild limbic (Racine 0: staring (str), 1–2: chewing and head-waving (C-HW)) than convulsive seizures (Racine 3: forelimb clonus (FLC), 4: rearing, 5: falling) overall (**p<0.01 between each seizure behavior with post-hoc Newman-Keuls pairwise comparisons, vehicle: n=12, blue; fadrozole: n=9, orange). Fadrozole reduced the cross-correlation for some types of seizures (*p<0.05 between vehicle- and fadrozole-treated rats, unpaired *t*-tests). (F–H) Mean ± SEM percent time spent in seizures plotted in 5 min bins for all behaviors more severe than staring (Racine 1–5, F), mild limbic (Racine 1–2, G) or convulsive (Racine 3–5, H) seizures showing that behavioral seizure escalation evident in vehicle-treated rats was inhibited by fadrozole (*p<0.05 between vehicle- and fadrozole-treated rats, post-hoc unpaired *t*-tests). (I) Percentages of vehicle (blue, left)- and fadrozole (orange, right)-treated rats in behavioral SE are shown in 5 min bins. Fadrozole decreased the incidence of behavioral SE throughout 2 hrs (*p<0.05, ANOVA). (J) Percentages of vehicle- and fadrozole-treated rats in electrographic SE, which occurred only during the 2nd hr, are shown in 5 min bins. Fadrozole decreased the incidence of electrographic SE during the 2nd hr (*p<0.05, post-hoc unpaired *t*-tests).

Behavioral seizures induced by KA followed a similar pattern as electrographic seizures in that vehicle-treated rats spent progressively more time in seizure, peaking during the 2nd hr when they spent $47 \pm 9\%$ of time in seizures more severe than staring (Racine stages 1–5, *Figure 2F*). This progressive increase was largely absent in fadrozole-treated rats, who spent only $16 \pm 8\%$ of time in Racine 1–5 seizures during the 2nd hr (time x drug: $F_{1,19}=7.22$, p<0.05, *Figure 2F*). The same results were obtained regardless of the types of seizures analyzed. For example, consideration of mild limbic seizures specifically (Racine 1–2, time x drug: $F_{1,19}=7.29$, p<0.05, *Figure 2G*) produced the same results, as did inclusion of staring (Racine 0–5, time x drug: $F_{1,19}=15.65$, p<0.01, not shown). Furthermore, in fadrozole-treated rats, convulsive seizures (Racine 3–5) typically subsided within 2 hrs, such that fadrozole-treated rats spent only ~3% of time in convulsive seizures, compared to 12% in vehicle-treated rats (time x drug: $F_{1,19}=6.59$, p<0.05, *Figure 2H*). Thus, inhibition of extra-gonadal aromatase suppressed behavioral seizures in parallel with its effect to suppress electrographic seizure activity in the hippocampus.

We also investigated how fadrozole affected SE, which is defined clinically as a prolonged seizure lasting 5 min or longer in the case of convulsive SE (*Trinka et al., 2015*). We analyzed KA-induced SE in two ways: based on sustained periods of either convulsive seizures (Racine 3–5) or of EEG amplitude exceeding 10x baseline (see Materials and methods). Both of these analyses showed that fadrozole suppressed SE. For example, fadrozole significantly decreased the proportion of rats in behavioral SE at any point during 2 hrs of testing ($F_{1,19}=7.66$, p<0.05, *Figure 2I*), reflecting decreased incidence of SE, and among rats that reached criteria for behavioral SE, the time spent in SE was also significantly lower in fadrozole-treated animals (Veh: $1296 \pm 245$ sec; Fad: $436 \pm 141$ sec, $t_{14}=2.26$, p<0.05, not shown). The latency to SE onset was longer on average in fadrozole-treated rats, but this was not a significant difference (Veh: $3052 \pm 262$ sec; Fad: $3548 \pm 520$ sec, p>0.10). Results were similar when SE was defined based on EEG amplitude, which met electrographic SE criteria only during the 2nd hr of testing. Fadrozole also significantly reduced the proportion of rats in electrographic SE during the 2nd hr ($F_{3,57}=7.38$, p<0.001, *Figure 2J*). These results showed that the seizure-suppressing effects of acute aromatase inhibition extend to SE.

## Neurosteroid E2 synthesis in the hippocampus

That fadrozole suppressed KA-induced seizures in gonadectomized animals showed that extra-gonadal aromatase activity, possibly in the brain, acutely modulates seizures. The hippocampus is a likely brain region in which aromatase activity could affect seizures, because of its relationship to the early stages of KA seizure activity, in vitro demonstrations of neurosteroid E2 synthesis, and many examples of acute E2 potentiation of neural activity in the hippocampus. However, there is currently no direct evidence for hippocampal estrogen synthesis in vivo. Therefore, we used in vivo microdialysis to investigate whether the hippocampus in awake, freely moving rats can synthesize E2. To focus specifically on E2 synthesized in the brain, animals were gonadectomized.

Parameters for microdialysis were first established in vitro. After optimization, we obtained $30.1 \pm 2.9\%$ of external E2 concentration recovered in dialysate (i.e., probe recovery) with little difference between concentrations of E2 ($31.3 \pm 5.4\%$ and $28.9 \pm 3.6\%$ for 0.25 and 1.0 ng/ml E2, respectively) or 1st and 2nd 30 min samples at each concentration ($25.9 \pm 2.1\%$ and $31.8 \pm 3.4\%$, respectively, *Figure 3A*). The latter result indicates that E2 concentration in dialysate reached a stable level within 30 min. Probe recovery was linear, evidenced by dialysate E2 concentration changing linearly with bath concentration (*Figure 3B*). We observed some carryover in the first post-E2 sample with both E2 concentrations ($31.0 \pm 0.3\%$ and $23.4 \pm 7.2\%$, for 0.25 and 1.0 ng/ml E2, respectively), but this was not detectable after 30 min (*Figure 3C*). In vivo tests showed that exogenously administered E2 (10 μg, s.c.) could be detected in the hippocampus within 30 min and then declined gradually over 2 hrs (*Figure 3D*).

To investigate neurosteroid E2 synthesis in vivo, we retrodialyzed androstenedione (4-dione, 1 ng/ml), an indirect precursor of E2 (*Figure 3—figure supplement 1A*), into the dorsal hippocampus. As shown in *Figure 3E*, 4-dione retrodialysis produced a large increase in local E2 concentration ($+8.9 \pm 3.2$ pg/ml or $310 \pm 67\%$ of baseline; $t_9=3.15$, p<0.05), demonstrating that the hippocampus can synthesize E2 in vivo. We confirmed that the enzyme immunoassay used to measure E2 (*Remage-Healey et al., 2008*) shows little to no cross-reactivity with steroids other than estrogens (*Figure 3—figure supplement 1B*). Cross-reactivity with 4-dione was 0.01% at 200 ng/ml, so the 1 ng/ml 4-dione dose used for retrodialysis would not be detected in the assay.

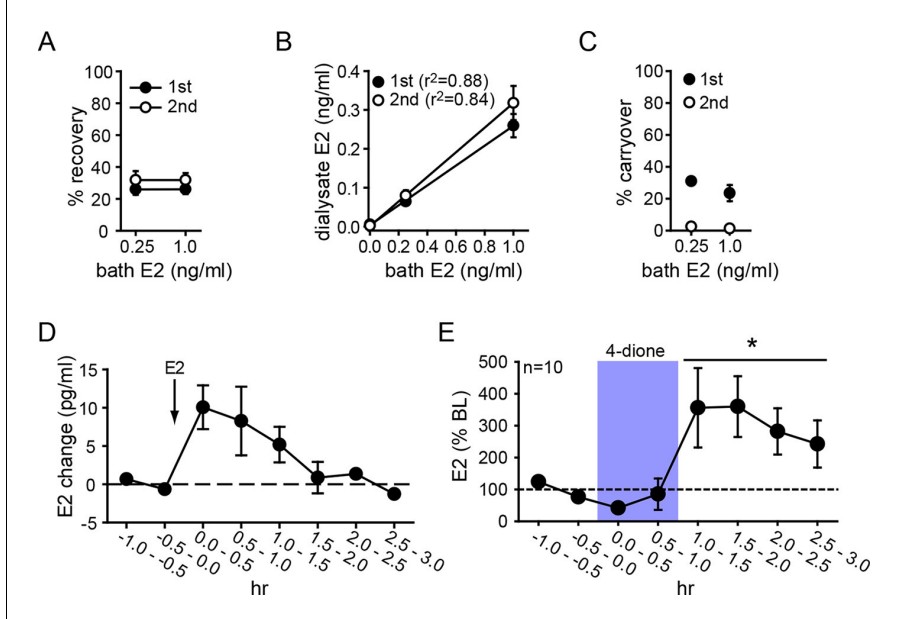

**Figure 3.** Microdialysis controls and basal neurosteroid E2 synthesis in the hippocampus. (**A**) Probe recovery was similar (~30%) at both E2 concentrations tested and in both 1st (closed circle) and 2nd (open circle) 30 min samples, indicating a fast response in dialysate to changes in external E2 concentration (n=6 probes). (**B**) Dialysate and bath E2 concentrations were linearly correlated, with little difference between the 1st (closed circles) and the 2nd (open circles) 30 min samples at each concentration. (**C**) Carryover of E2 was evident in the 1st (closed circle) but not in the 2nd (open circle) 30 min samples after transferring a probe from an E2-containing to an E2-free bath. (**D**) E2 was detected in the hippocampus within 30 min of systemic E2 injection (10 μg in 25% ethanol/saline, s.c., n=2), followed by a gradual decline over 2 hr. (**E**) Retrodialysis of androstenedione (4-dione, 1 ng/ml, blue box), increased hippocampal E2 concentration by +8.9 ± 3.2 pg/ml (310 ± 67%, n=10, *p<0.05 relative to baseline (BL), paired *t*-tests).

The following figure supplement is available for figure 3:

**Figure supplement 1.** Synthetic pathway for C19 and C18 steroids and the results of cross-reactivity/interference testing for EIA.

## Seizures stimulate neurosteroid E2 synthesis in the hippocampus

As noted previously, in vitro studies suggest a positive feedback loop between neuronal activity and E2 synthesis in the hippocampus. As such, seizures in vivo might stimulate E2 synthesis, which could in turn potentiate further seizure activity. We investigated whether acute seizures affect hippocampal E2 levels using in vivo microdialysis (*Figure 4—figure supplement 1*) in gonadectomized rats treated with KA (1.25–5 mg/kg, i.p.). This showed that, overall, E2 levels in the hippocampus more than doubled during a 2 hr period after KA ($F_{2,60}$=9.15, p<0.001, *Figure 4A*). The mean basal concentration of E2 we detected in dialysate was 6.3 ± 0.8 pg/ml (23.3 ± 2.9 pM); levels were slightly higher in females (7.1 ± 1.0 pg/ml) than in males (5.4 ± 1.2 pg/ml), but the difference was not statistically significant ($t_{41}$=1.10. p=0.28). The highest E2 concentrations we measured were 16.7 pg/ml (61.3 pM) during baseline and 30.0 pg/ml (110.1 pM) after KA injection. The seizure-induced increase in hippocampal E2 was similar in both sexes and hemispheres (*Figure 4—figure supplement 2*).

The severity of seizures induced by KA varies among individual animals, which allowed us to compare seizure severity with changes in hippocampal E2 levels. We divided animals into those that experienced mild, moderate, or severe seizures based on the amount of time that each animal spent in seizure behaviors more severe than staring (Racine 1–5) during 2 hrs after KA. This demonstrated that animals with relatively mild seizures (<30 min, n=10, *Figure 4B*) showed a small increase in E2 (188 ± 38% baseline, $F_{2,18}$=5.39, p<0.05) that was delayed relative to KA administration, whereas

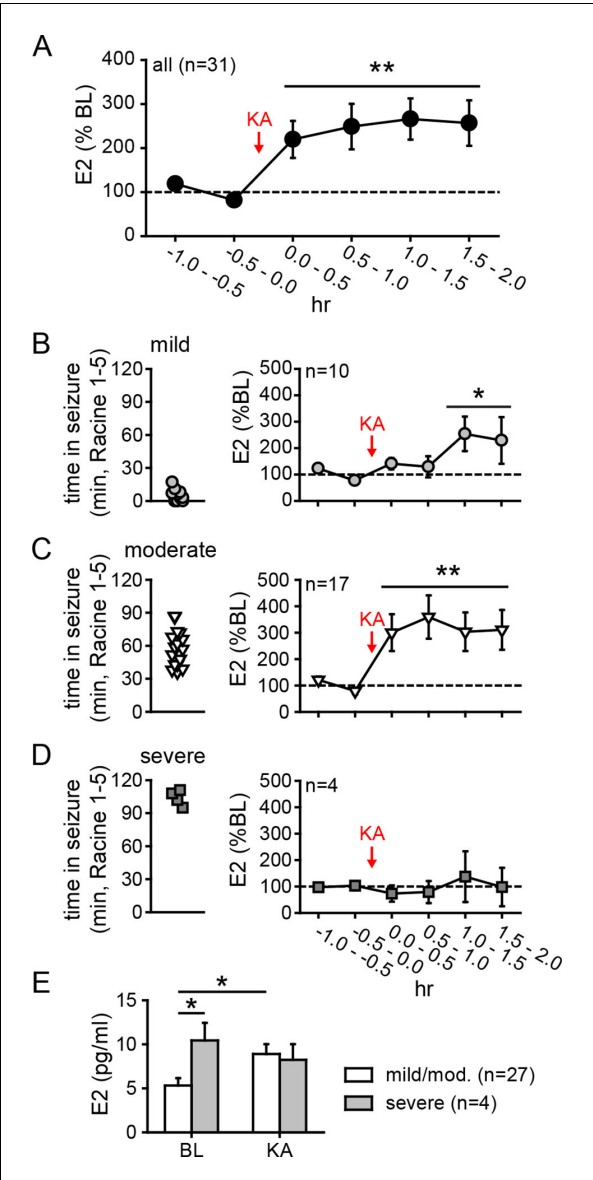

**Figure 4.** Hippocampal neurosteroid E2 levels increase during KA seizures. (**A**) Mean ± SEM percent baseline (BL) E2 before and after KA showing that KA-induced seizures increased hippocampal E2 concentration (\*\*p<0.001 relative to BL, post-hoc paired *t*-tests). (**B-D**) Time in Racine 1–5 seizures (left) was used to classify rats into those experiencing mild (<30 min, **B**), moderate (30–90 min, **C**), or severe (>90 min, **D**) seizures, and E2 levels (right) were measured before and after KA. Rats with mild seizures (**B**) showed a small delayed increase in hippocampal E2, whereas rats with moderate seizures (**C**) showed a large (x3 BL) increase within 30 min (\*p<0.05, \*\*<0.01 relative to BL, paired post-hoc *t*-tests). E2 levels did not change in rats with the most severe seizures (**D**). (**E**) Mean ± SEM E2 concentrations during baseline (BL) and after KA injection (KA) for rats with mild/moderate vs. severe seizures shows that basal E2 was significantly higher in rats in which KA produced severe seizures (\*p<0.05, *t*-test), whereas E2 increased from low to high levels in rats in which KA produced mild/moderate seizures (\*p<0.05, *t*-test).

The following figure supplements are available for figure 4:

**Figure supplement 1.** Placements of microdialysis probes.

**Figure supplement 2.** No difference in seizure-induced changes in hippocampal E2 by sex or hemisphere.

those with moderate seizures (30–90 min, n=17, *Figure 4C*) showed a large increase in E2 (318 ± 68% baseline, $F_{2,23}$=8.22, p<0.01) evident within the first 30 min. Interestingly, the small fraction of animals with very severe seizures (>90 min, n=4, *Figure 4D*) began with significantly higher basal E2 (10.4 ± 2.0 pg/ml) than those with mild or moderate seizures (5.3 ± 0.8 pg/ml, $t_{29}$=2.22, p<0.05, *Figure 4E*) and hippocampal E2 remained high during seizures in those animals (97 ± 33% baseline). This is in contrast to animals with mild or moderate seizures in which E2 levels increased to 8.9 ± 0.8 pg/ml, a significant difference from their baseline ($t_{52}$=2.54, p<0.05, *Figure 4E*). That already high hippocampal E2 did not increase further in animals with the most severe seizures suggests a ceiling on hippocampal E2 and raises the possibility that high basal E2 in the hippocampus of a small number of animals may have predisposed them to more severe seizures. Together, the results of microdialysis experiments demonstrated that KA seizures increase E2 levels in the hippocampus and indicated that high hippocampal E2 is associated with greater seizure severity.

## Inhibition of hippocampal E2 synthesis suppresses electrographic seizures

The results with systemic inhibition of aromatase demonstrated that extra-gonadal estrogen synthesis during seizures contributes to the escalation of seizures from mild to severe. Coupled with results from microdialysis showing seizure-induced E2 synthesis in the hippocampus, this suggests that E2 synthesized in the hippocampus may be acutely seizure-promoting. To test this possibility, we first bilaterally infused fadrozole (10 μg/side, n=10) or vehicle (30% β-cyclodextrin in saline, n=11) into the dorsal hippocampus 20 min before KA (10 mg/kg, i.p.) and monitored seizures for 2 hrs. We used pre-treatment in this experiment because of difficulties associated with intra-hippocampal infusion after the onset of seizures and to avoid the possibility that infusions would disrupt EEG recording. *Figure 5A* shows data from a representative vehicle-infused animal, demonstrating the same pattern of hippocampal electrographic activity and accompanying behavioral seizures as in non-infused animals (*Figure 1*, *2*). Also as seen previously, increased EEG activity in the hippocampus often corresponded to behaviors characteristic of mild limbic seizures (*Figure 5B*) and the relationship between electrographic seizure activity was stronger for mild limbic than convulsive seizure behaviors (amplitude: $F_{4,94}$=44.02, p<0.001, *Figure 5C*; power: $F_{4,94}$=21.13, p<0.001, *Figure 5D*). In contrast to systemic fadrozole, however, intra-hippocampal fadrozole did not affect the strength of cross-correlation between EEG amplitude (*Figure 5C*) or power (*Figure 5D*) and seizure behaviors.

Importantly, among rats that exhibited electrographic seizures, fadrozole significantly attenuated the seizure-related increase in β-low γ power ($t_{19}$=2.43, p<0.05, *Figure 5E,F*), showing that intra-hippocampal fadrozole suppressed electrographic seizure activity. In contrast to systemic inhibition of aromatase with fadrozole (*Figure 1*), the effect of intra-hippocampal fadrozole was specific to the β-low γ frequency range; neither δ-θ (1–10 Hz) nor ripple (100–200 Hz) power showed significantly different seizure-related changes (p values >0.2, *Figure 5E*). Time spent in seizures as detected by 5x baseline power in the β-low γ range was significantly reduced by intra-hippocampal fadrozole, however ($t_{19}$=2.62, p<0.05; *Figure 5G,H*). Results were similar with 3x and 10x baseline thresholds (3x: $t_{19}$=2.94, p<0.01, 10x: $t_{19}$=2.21, p<0.05, not shown). These findings demonstrated that inhibiting aromatase activity selectively in the hippocampus suppresses electrographic seizure activity induced by KA. Differences in the effects of systemic vs. intra-hippocampal fadrozole, e.g., a stronger effect of systemic fadrozole on time spent in convulsive seizures and broader effects on seizure-induced increases in EEG power, suggest that synthesis of neurosteroid estrogens in brain regions in addition to the hippocampus also may be important in seizures.

## Inhibition of hippocampal E2 synthesis suppresses behavioral seizures

One concern in experiments with EEG recording is the possibility that tissue damage due to EEG electrodes could influence seizure susceptibility and/or aromatase activity or expression, thus altering the effect of aromatase inhibition. For example, previous studies have shown that brain injury can induce aromatase expression in glia (*Peterson et al., 2001*; *Carswell et al., 2005*). To confirm that the effect of fadrozole to suppress seizures was not influenced by implanted electrodes, we repeated KA treatment (15 mg/kg) with intra-hippocampal vehicle (n=29) or fadrozole (n=29) in animals without electrodes in the hippocampus. The dose of KA was increased to achieve the same overall levels of seizure severity as in animals with EEG electrodes implanted. As in previous

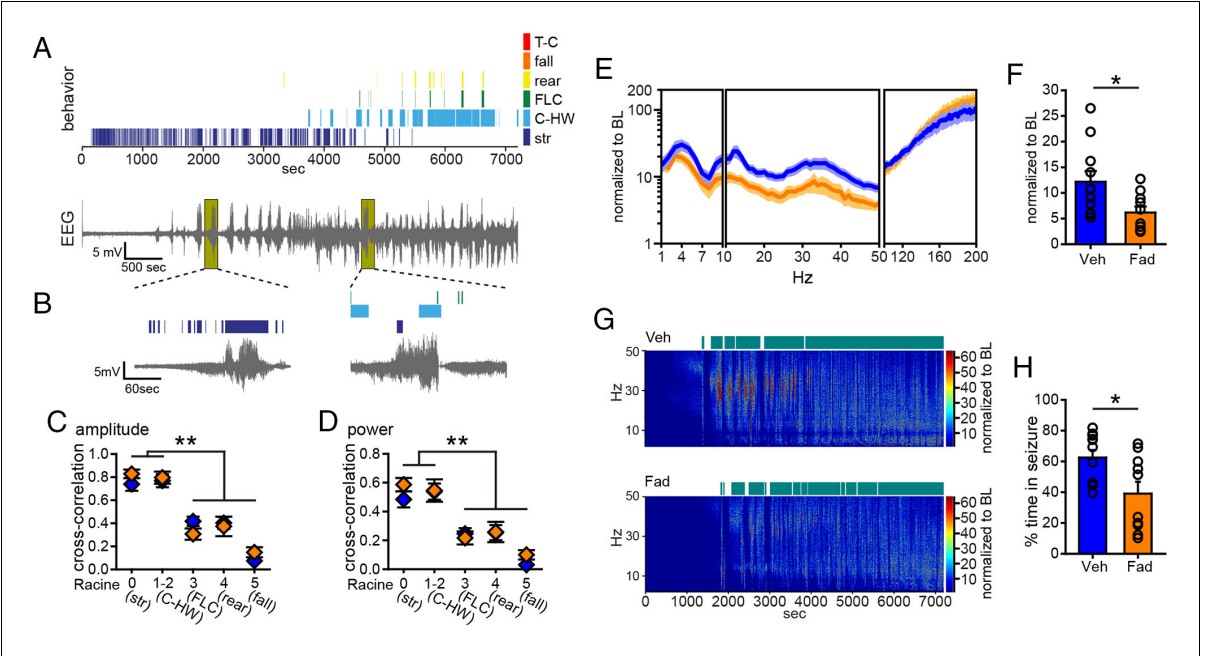

**Figure 5.** Acute intra-hippocampal aromatase inhibition suppresses electrographic seizures in gonadectomized rats. (**A**) Representative time-synched behavioral (top) and electrographic seizures (bottom) following KA. (**B**) Magnified views of early (left) and middle (right) electrographic seizures and accompanying seizure behaviors. (**C**, **D**) Mild limbic seizures (Racine 0: staring (str), and 1–2: chewing and head-waving (C-HW)) showed higher correlation with normalized hippocampal EEG amplitude (**C**) and β-low γ power (**D**) than convulsive seizure behaviors (Racine 3: forelimb clonus (FLC), 4: rearing, 5: falling) (**p<0.001 between each behavior, post-hoc Newman-Keuls pairwise comparisons, vehicle: n=11, blue; fadrozole: n=10, orange). Intra-hippocampal infusion of fadrozole did not alter these relationships. (**E**) Mean ± SEM normalized power spectrogram plotted in 1 Hz bins for δ-θ (1–10 Hz), β-low γ (10–50 Hz), and ripple (100–200 Hz) frequency ranges for vehicle and fadrozole rats showing the seizure-induced increase in power in the β-low γ range in both groups. (**F**) Fadrozole attenuated the seizure-induced increase in power in the β-low γ range (*p<0.05, unpaired *t*-test) but not the δ-θ or ripple ranges. (**G**) Representative normalized power spectrum from one vehicle- (top) and one fadrozole- (bottom) treated rat. Teal bars above heat maps indicate seizures detected by 5x baseline power in the β-low γ range as the threshold. (**H**) Fadrozole decreased time that power in the β-low γ range exceeded 5x baseline (*p<0.05, unpaired *t*-test).

experiments, seizures were monitored for 2 hrs after KA. This showed that fadrozole significantly delayed the onset of behavioral seizures (Racine 0, $t_{52}$=2.57, p<0.05, *Figure 6A*) and strongly attenuated seizure progression. While the proportion of rats exhibiting the earliest stage of behavioral seizure, staring (Racine 0), was the same in vehicle- and fadrozole-infused animals, significantly fewer fadrozole-treated rats progressed beyond staring and the majority never progressed beyond mild limbic seizures. This was in contrast to vehicle-treated rats in which roughly 2/3 progressed to convulsive seizures ($\chi^2$-tests, p<0.05 or p<0.01, *Figure 6B*). In addition, even among rats that progressed to more severe seizure behaviors, fadrozole significantly reduced the time animals spent in any seizures beyond staring (Racine 1–5, time x drug: $F_{1,30}$=6.99, p<0.05; *Figure 6C*) and in mild limbic seizures specifically (Racine 1–2, time x drug: $F_{1,30}$=5.64, p<0.05, *Figure 6D*) in the 2nd hr. Mean time spent in convulsive seizures (Racine 3–5) was lower in fadrozole-treated rats, but this was not a significant difference (time x drug: $F_{1,24}$=0.74, p=0.40). Seizure suppression by intra-hippocampal fadrozole is also readily visible in heatmaps plotting the maximum seizure stage reached per minute in individual animals (*Figure 6—figure supplement 1*). These results showed that the seizure-suppressing effect of acutely inhibiting hippocampal aromatase activity is not related to tissue damage resulting from hippocampal electrodes.

The ability of fadrozole to suppress seizures induced by KA suggests that acute administration of aromatase inhibitors may be a treatment for SE. For this to be feasible, however, fadrozole should not exacerbate seizure-induced cell death. Longer-term treatment with E2 is known to be neuroprotective, including in the KA seizure model (*Hoffman et al., 2003*; *Schauwecker et al., 2009*). We tested whether fadrozole affected delayed hippocampal cell death after KA using Fluoro-Jade B staining. This showed no differences between vehicle- and fadrozole-treated animals ($t_{21}$<0.45,

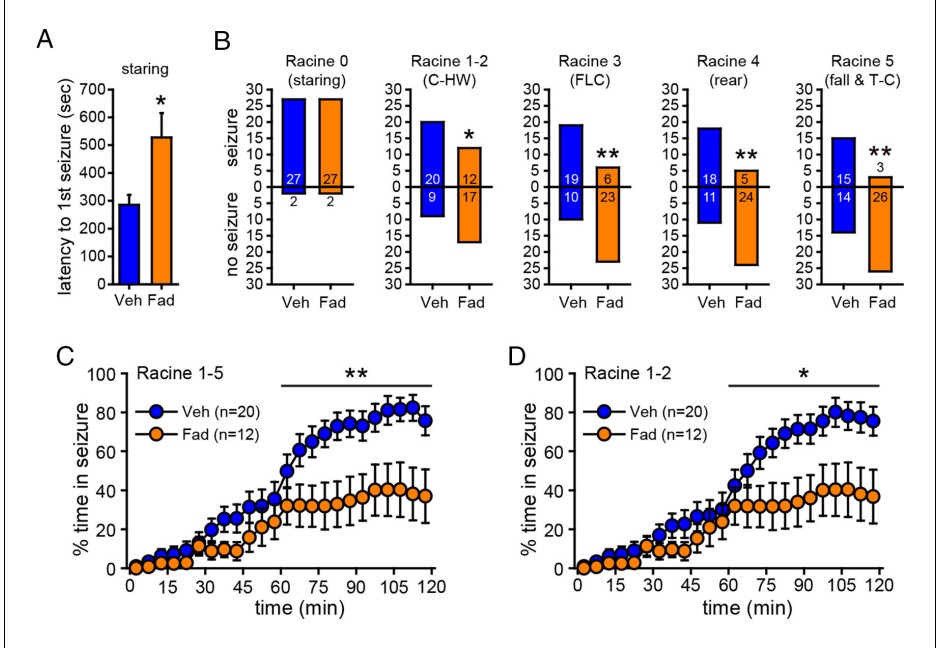

**Figure 6.** Acute intra-hippocampal aromatase inhibition delays seizure onset and suppresses behavioral seizures in gonadectomized rats. (**A**) Fadrozole (Fad, n=29) increased latency to the 1st occurrence of staring compared to vehicle (Veh, n=29, **p<0.05, unpaired *t*-tests). (**B**) Fadrozole significantly reduced the proportions of rats exhibiting behavioral seizures beyond staring (*p<0.05 and **p<0.01 relative to Veh, $X^2$-tests). Among the rats that reached particular seizure stages, fadrozole reduced the time spent in those seizures during the 2nd hr, considering any seizure stage beyond staring (Racine 1–5, **C**) or mild limbic seizures (Racine 1–2, **D**) (*p<0.05, **p<0.01 compared to Veh, post-hoc unpaired *t*-tests).
The following figure supplements are available for figure 6:

**Figure supplement 1.** Behavioral seizures shown in individual rats.
**Figure supplement 2.** No effect of intra-hippocampal aromatase inhibition on seizure-induced cell death in the hippocampus of gonadectomized rats.

p>0.66, for all subregions, *Figure 6—figure supplement 2*), demonstrating that acute aromatase inhibition does not exacerbate seizure-induced cell death.

## Systemic aromatase inhibition acutely suppresses seizures in gonadally intact animals

Our experiments thus far utilized gonadectomized animals to eliminate the influence of gonadal hormones and focus on E2 synthesized in the brain. However, for aromatase inhibitors to be clinically useful, they should suppress seizures when gonads are intact, as in the majority of seizure patients. Therefore, we tested how systemic administration of an aromatase inhibitor that is used clinically, letrozole (*Bhatnagar, 2007*), influences seizures in gonadally intact male and female rats. As in previous experiments (*Figures 1*, *2*), we injected vehicle or the aromatase inhibitor i.v. after the first electrographic seizure was detected in the hippocampus and seizures detected prior to i.v. injections did not differ in latency, duration, amplitude, or β-low γ power among rats in either treatment group (*Figure 7—figure supplement 1*). Seizure recording was extended to 6 hrs to better evaluate the effects of aromatase inhibition on SE (*Figure 7A*). Overall, behavioral seizures were more severe with gonads intact, consistent with a proconvulsant effect of circulating hormones (*Terasawa and Timiras, 1968*; *Nicoletti et al., 1985*; *Woolley, 2000*; *Mejias-Aponte et al., 2002*). This was particularly evident in females, in which it was not feasible to use 10 mg/kg KA. Thus we lowered the dose of KA to 7.5 mg/kg in females (n=5 vehicle, n=4 letrozole), to achieve the same range of seizure severity as in males treated with 10 mg/kg KA (n=6 vehicle, n=6 letrozole). Statistical analyses in gonadally intact animals therefore included KA dose as a covariate.

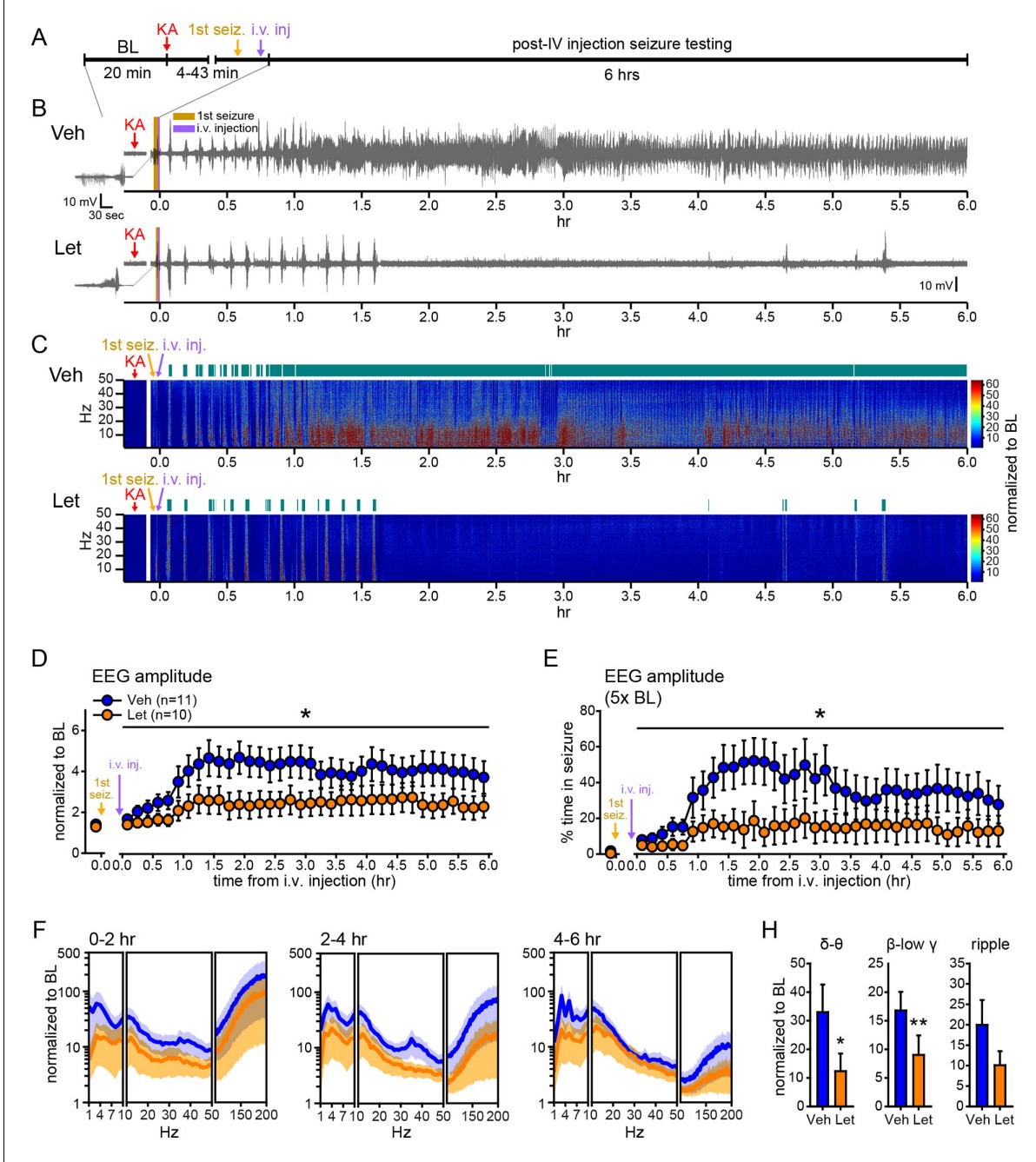

**Figure 7.** Systemic inhibition of aromatase after seizure onset suppresses electrographic seizures in gonadally intact rats. (**A**) Schematic of experimental design. (**B**, **C**) Representative EEG (**B**) and normalized power spectrum (**C**) from one vehicle (Veh, top)- and one letrozole (Let, bottom)-treated rat. For EEG, the initial seizure in each animal is shown on the left. Teal bars above heatmaps indicate seizures detected by 5x baseline power in the β-low γ (10–50 Hz) range as the threshold. (**D**) Mean ± SEM normalized EEG amplitude plotted in 10 min bins for vehicle (blue, n=11)- and letrozole (orange, n=10)-treated rats. The seizure-related increase in EEG amplitude was suppressed by letrozole throughout 6 hrs of recording (p<0.05 post-hoc unpaired *t*-tests). (**E**) Mean ± SEM percent time in seizure using 5x baseline thresholds of EEG amplitude showing that the progressive increase in time in seizure evident in vehicle-treated rats was suppressed by letrozole (p<0.05 post-hoc unpaired *t*-tests). (**F**) Mean ± SEM normalized power spectrogram plotted in 1 Hz bins for δ-θ (1–10 Hz), β-low γ (10–50 Hz), and ripple (100–200 Hz) frequency ranges for vehicle (blue)- and letrozole (orange)-treated rats are shown for each 2 hr epoch. (**H**) Seizure-induced increases in power relative to baseline were lower in letrozole- than vehicle-treated rats in δ-θ and β-low γ ranges (ANCOVA, *p<0.05 and **p<0.01 between vehicle- and letrozole-treated rats, post-hoc unpaired *t*-tests). BL = baseline.

The following figure supplement is available for figure 7:

Figure 7 continued

**Figure supplement 1.** No difference in characteristics of 1$^{st}$ seizures in gonadally intact rats subsequently treated with vehicle or letrozole.

*Figure 7B and C* show representative recordings of electrographic seizures in one vehicle- and one letrozole-treated rat throughout 6 hrs of seizure monitoring. Analysis of these seizures showed that letrozole significantly reduced the seizure-related increase in EEG amplitude, by 40% ($F_{1,18}$=6.63, p<0.05, *Figure 7D*), and decreased the time spent in seizures based on a 5x baseline threshold of EEG amplitude, by 61% ($F_{1,18}$=4.75, p<0.05, *Figure 7E*), throughout all 6 hrs of recording. Examination of changes in EEG power in 2 hr epochs (*Figure 7F*) showed that letrozole also attenuated the seizure-induced increase in power across all 6 hrs of recording. The effect of letrozole was statistically significant in the δ-θ ($F_{1,18}$=4.56, p<0.05) and β-low γ ranges ($F_{1,18}$=5.54, p<0.05) but not in the ripple range ($F_{1,18}$=2.82, p=0.11) (*Figure 7H*). Time spent in seizures as assessed by a 5x baseline increase in power in β-low γ range was also significantly lower in rats treated with letrozole ($F_{1,18}$=8.15, p<0.05, not shown). Thus the seizure-suppressing effect of aromatase inhibition evident in gonadectomized animals is also observed in gondally intact animals and seizure suppression by aromatase inhibitors persists through at least 6 hrs after seizure initiation. These results support the idea that aromatase inhibitors may be clinically useful for acute seizure suppression.

We also investigated how acute aromatase inhibition affects SE in gonadally intact animals. As with gonadectomized rats (*Figure 2I,J*), we analyzed SE in two ways: based on sustained periods of either frequent convulsive seizures (Racine 3–5; *Figure 8A*) or EEG amplitude that exceeded 10x baseline (*Figure 8B*). Both analyses showed that letrozole suppressed SE in gonadally intact rats. The proportion of rats in behavioral SE was significantly lower in letrozole- than vehicle-treated animals throughout 6 hrs of testing ($F_{1,18}$=5.13, p<0.05, *Figure 8C*), as was the proportion of rats in electrographic SE ($F_{1,18}$=7.85, p<0.05, *Figure 8D*). These results confirmed that the effects of aromatase inhibition to suppress seizure activity in gonadally intact rats translate to decreased incidence of SE. Among rats in each group that reached behavioral SE, average time spent in SE was less than half in letrozole- compared to vehicle-treated rats (Veh: 7278 ± 1770 sec, Let: 3213 ± 1582 sec) but this was not a statistically significant difference ($t_{17}$=1.70, p=0.11). Similarly, the average latency to behavioral SE onset was longer in letrozole- than vehicle-treated rats (Veh: 2085 ± 387 sec, Let: 3131 ± 640 sec), but the difference was not significant ($t_{17}$=1.43, p=0.17).

The impact of gonadal status on seizure severity noted above is evident when comparing vehicle-treated gonadectomized (n=5) and gonadally intact males (n=6), both of which received the same 10 mg/kg dose of KA and were otherwise treated identically. Intact males spent twofold more time in Racine 1–5 seizures in the first 2 hrs after KA than gonadectomized males did ($F_{1,9}$=15.67, p<0.01, *Figure 8—figure supplement 1A*). Behavioral seizures also began much more quickly in gonadally intact males. Racine 1–2 seizures began within 231 ± 112 s in intact males compared to 3193 ± 1251 s in gonadectomized males ($t_9$=2.61, p<0.05) and all 6 intact males reached Racine 3–5 seizures within 1 hr after KA whereas only 1 of 5 gonadectomized males did ($\chi^2$=7.54, p<0.05). Interestingly, while the difference between gonadectomized and intact males was very clear in behavioral seizures, this difference was not evident in EEG recordings. For example, normalized EEG amplitude increased after KA with a similar time course in both groups (p>0.50, *Figure 8—figure supplement 1B*).

Finally, we investigated whether systemic aromatase inhibition with letrozole influenced delayed hippocampal cell death after KA in gonadally intact animals using Fluoro-Jade B staining, as we did following intra-hippocampal fadrozole in gonadectomized animals. We used the 6 vehicle- and 6 letrozole-treated males for this analysis because males had received the same dose of KA as in our cell death analysis after intra-hippocampal fadrozole. Counts of Fluoro-Jade B-positive cells showed no significant differences between vehicle and letrozole groups in any hippocampal subregion examined (p>0.19, *Figure 8—figure supplement 2*). Thus, like intra-hippocampal aromatase inhibition, systemic aromatase inhibition does not appear to exacerbate seizure-induced cell death. The degree of cell death evident in CA3 was most closely related to seizure severity in that the 4 of 6 letrozole-treated rats that did not experience electrographic SE during testing had significantly fewer Fluoro-Jade B-positive cells in CA3 than the 2 of 6 letrozole-treated and all 6 vehicle-treated rats that did reach SE ($t_{10}$=2.36, p<0.05).

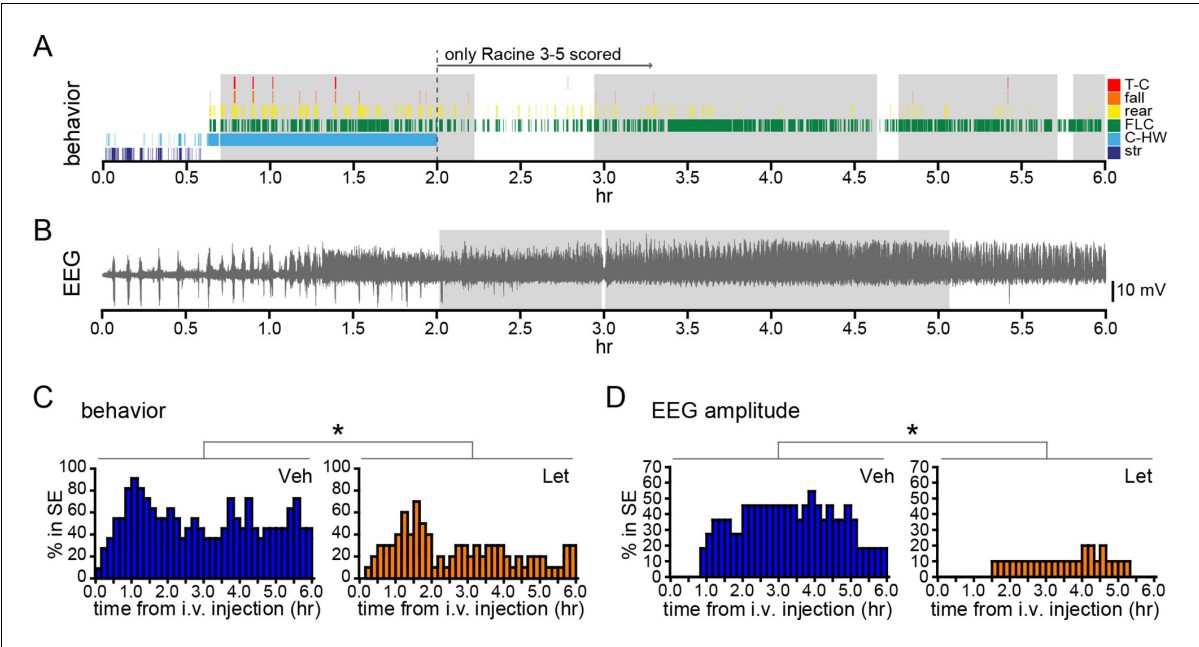

**Figure 8.** Systemic inhibition of aromatase after seizure onset suppresses status epilepticus in gonadally intact rats. Behavioral (**A**) and electrographic (**B**) seizures from a representative vehicle-treated rat with periods of SE (status epilepticus) highlighted in gray. All Racine 0–5 seizure behaviors were scored for the first 2 hrs whereas only Racine 3–5 seizure behaviors were scored for the remaining 4 hrs. (**C**) Percentages of vehicle (blue, left, n=11)- and letrozole (orange, right, n=10)-treated rats in behavioral SE plotted in 10 min bins. Letrozole decreased the incidence of behavioral SE (*p<0.05, ANCOVA). (**D**) Percentages of the same vehicle- and letrozole-treated rats in electrographic SE plottedin 10 min bins. Letrozole decreased the incidence of electrographic SE (*p<0.05, ANCOVA).

The following figure supplements are available for figure 8:

**Figure supplement 1.** Castration attenuated behavioral, but not electrographic seizures.

**Figure supplement 2.** No effect of systemic aromatase inhibition on seizure-induced cell death in the hippocampus of gonadally intact rats.

## Discussion

These results reveal neurosteroid estrogen synthesis as a previously unknown factor in the progressive escalation of seizures in an animal model of SE and provide the first in vivo evidence of a functional role for neurosteroid estrogens in the hippocampus. We found that systemic administration of an aromatase inhibitor after the onset of KA seizures strongly suppressed electrographic and behavioral seizure activity, in both sexes, without additional interventions. Microdialysis showed that seizures stimulate *de novo* synthesis of estrogens in the hippocampus and that animals with more severe seizures began with or reached higher levels of E2 in the hippocampus. Because E2 is known to acutely promote hippocampal neural activity, this suggested that E2 synthesized in the hippocampus might be seizure-promoting. Consistent with this, infusion of an aromatase inhibitor specifically into the hippocampus also suppressed both electrographic and behavioral seizures. These results support the idea of a positive feedback loop between seizure activity and neurosteroid estrogens in which seizures stimulate estrogen synthesis in the brain, which then acutely promotes neural activity, contributing to further seizure activity. Breaking this cycle with aromatase inhibitor therapy may be a novel approach to clinical control of SE.

### Mechanisms by which estrogens acutely modulate hippocampal activity

Decades of in vitro studies have shown that E2 acutely potentiates neural activity in the hippocampus. Since the early work of *Teyler et al. (1980)*, which showed that E2 application to hippocampal slices rapidly increases extracellularly recorded field potentials, multiple groups have demonstrated acute E2 modulation of hippocampal intrinsic and synaptic physiology. For example, E2 suppresses

$Ca^{++}$-activated $K^+$ currents to reduce the slow afterhyperpolarization following a burst of action potentials (*Kumar and Foster, 2002*; *Carrer et al., 2003*) and increases repetitive action potential firing in response to current injection (*Wu et al., 2011*). E2 also acutely promotes excitatory synaptic transmission through both increased glutamate release probability and increased postsynaptic sensitivity to glutamate (*Kramar et al., 2009*; *Smejkalova and Woolley, 2010*; *Oberlander and Woolley, 2016*). The molecular mechanisms that underlie these effects are under active investigation by our lab and others. Most studies indicate that E2's acute actions are mediated by membrane-associated forms of the classical estrogen receptors, ERα (*Huang and Woolley, 2012*; *Tabatadze et al., 2015*) and ERβ (*Kramar et al., 2009*; *Smejkalova and Woolley, 2010*), as well as the G protein-coupled ER, GPER1 (*Kumar et al., 2015*; *Oberlander and Woolley, 2016*), and involve downstream signaling through protein kinases including Src, Erk1/2, protein kinase A, and RhoA kinase (*Boulware et al., 2005*; *Kramar et al., 2009*; *Zadran et al., 2009*) to influence synaptic transmission. Although it is possible that inhibiting aromatase resulted in increased levels of estrogen precursors, such as testosterone or androstenedione, the well-known effects of E2 to acutely promote neural activity make an estrogen-based explanation for the seizure-suppressing effects of aromatase inhibitors the most parsimonious. Further studies will be necessary to investigate the possibility that aromatase inhibition suppresses seizures by influencing other steroids in addition to E2.

Our microdialysis results indicate that E2 reaches sufficient concentrations in the hippocampus to promote seizure activity. For example, based on our probe recovery estimate of ~30% and the mean basal concentrations of E2 we detected in dialysate, basal E2 concentrations in the hippocampus of gonadectomized rats average at least ~20 pg/ml (74 pM). The highest concentrations we measured indicate that hippocampal E2 reaches ~100 pg/ml (367 pM) after KA, which overlaps with E2 concentrations known to acutely potentiate intrinsic excitability and synaptic transmission when bath-applied in vitro (100 pM, e.g., *Smejkalova and Woolley, 2010*; *Wu et al., 2011*). Separately, two considerations suggest that our estimates of extracellular E2 levels are likely to underestimate in vivo concentrations at or near synapses, where extranuclear ERs are located (*Milner et al., 2001*; *2005*; *Hart et al., 2007*; *Waters et al., 2015*). First, electron microscopic immunogold labeling indicates that aromatase is concentrated at synapses (*Hojo et al., 2004*). This suggests that synaptic E2 levels could be substantially higher than levels measured using microdialysis, which samples a large extracellular volume. Second, we obtained the in vitro probe recovery value of 30% with no E2 carrier (e.g., BSA) in the bath; this value therefore reflects maximum recovery. If E2 is carrier-bound in the brain, then total E2 levels could be substantially higher than our estimates indicate. These considerations make it very likely that E2 reaches sufficient concentrations in the brain during seizures to potentiate neuronal excitability and/or synaptic transmission and thereby promote further seizure activity through one or more of the mechanisms that have been indentified in vitro.

## Regulation of aromatase activity in the brain

We have shown that seizures increase E2 levels in the hippocampus, but how would E2 eventually return to baseline? Passive diffusion, active reuptake, and/or active metabolism could be involved. Passive diffusion in extracellular space is possible, as the maximum solubility of E2 in aqueous medium is estimated to be in the low μM range (*Shareef et al., 2006*). In addition, the lipophilic nature of estrogens allows their diffusion across or within membranes, including the plasma membrane, which could reduce E2 levels in extracellular space. Sequestration and/or transport of locally synthesized E2 may also be aided by carrier molecules, as is the case for steroids in circulation. Steroids bound to sex hormone binding globulin (SHBG) can be actively transported across the plasma membrane in an immortalized hippocampal cell line (*Caldwell et al., 2007*), although it is currently not known whether SHBGs are expressed in the hippocampus in vivo.

Interestingly, there is some evidence that active metabolism of estrogens could occur with intracellular pH changes associated with seizures. Studies in peripheral systems show that the aromatase enzyme can switch between aromatase and hydroxylase activity in a pH-dependent manner (*Osawa et al., 1993*). Aromatization is the final step in E2 synthesis; hydroxylation is an initial step in conversion of E2 to an inactive form (*Lee et al., 2003*). The ratio of aromatase to hydroxylase activity is greatest at acidic pH, whereas hydroxylase activity predominates at alkaline pH. This is potentially relevant to seizures in that, in vitro, KCl-induced depolarization of neurons (*Svichar et al., 2011*) or treatment with the NMDA receptor co-agonist glycine (*Diering et al., 2011*) produces biphasic changes in intracellular pH: an initial acid transient is followed by alkalinization. Thus, the proximity

of aromatase to acid-loading or acid-extruding membrane proteins that mediate activity-dependent changes in intracellular pH during seizures (*Sinning et al., 2011*) may control a balance of aromatase vs. hydroxylase activity in a way that initially promotes E2 synthesis, followed by E2 inactivation through hydroxylation. Further studies will be required to test this idea.

It should be noted that several studies on avian aromatase have demonstrated mechanisms through which aromatase activity can be negatively regulated, but their results indicate effects that are the opposite of what is observed in the rodent hippocampus. For example, glutamate retrodialysis reduced E2 levels in a cortical brain area in zebra finches (*Remage-Healey et al., 2008*) and treatment of quail hypothalamic explants with glutamate receptor agonists lowered aromatase activity (*Balthazart et al., 2006*). In contrast, glutamate receptor activation increases aromatase activity in rat hippocampal slices (*Hojo et al., 2004*) and our findings with in vivo seizures in rats are consistent with this. Thus there may be species- and/or brain region-specific regulation of aromatase activity. One possibility is that species differences in enzyme structure lead to differences in aromatase regulation. Aromatase activity requires access to lipophilic substrate, and differences between avian and mammalian aromatase are greater in a region responsible for membrane association (~58% identity) than catalytic activity (~73% identity, chicken vs. rat, *Ghosh et al., 2009*).

That we detected a range of basal E2 concentrations suggests that factors in addition to seizures may regulate aromatase activity in the hippocampus. For example, studies in the avian brain indicate that stress regulates enzymes involved in E2 metabolism (*Soma et al., 2004*; *Dickens et al., 2011*). Given our observation that high basal E2 is associated with more severe seizures in response to KA, understanding how basal E2 levels in the hippocampus are influenced by stress or other experience may have important implications for understanding the role of environmental factors in modulating seizure susceptibility and/or severity.

### Therapeutic potential of acute aromatase inhibitor treatment

Although most of our experiments were done in gonadectomized animals to focus on E2 synthesized in the brain, additional experiments showed that systemic aromatase inhibition also acutely suppresses seizures when circulating gonadal hormones are present. Thus, acute aromatase inhibitor treatment may be useful in seizure patients, most of whom are gonadally intact. One concern, however, is that aromatase inhibition would eliminate the neuroprotective effects of estrogens (*Hoffman et al., 2003*; *Schauwecker et al., 2009*), which could have the detrimental effect of exacerbating seizure-related damage in the brain. We investigated this and found no effects of either intra-hippocampal fadrozole in gonadectomized animals or systemic letrozole in gonadally intact animals on seizure-induced cell death. It is possible that the milder seizures with aromatase inhibition, which would be expected to result in less cell death, were balanced by loss of estrogen neuroprotection producing roughly equivalent seizure-induced cell death. Further experiments with more detailed analyses of seizure-related neuronal damage will be required to investigate how aromatase inhibition interacts with estrogen neuroprotection to influence downstream sequelae of seizure activity.

## Materials and methods

### Animals and surgeries

All animal procedures were performed in accordance with the *National Institutes of Health Guide for the Care and Use of Laboratory Animals* and were approved by the Northwestern University Animal Care and Use Committee (Animal Study Protocol: IS00000520, Animal Welfare Assurance: A3283-01). Young adult male and female Sprague-Dawley rats (~50 days old, Harlan, Indianapolis, IN) were gonadectomized or gonadally intact, and were implanted with either unilateral microdialysis guide cannulae (CMA, Kista, Sweden) or bilateral intra-hippocampal infusion guide cannulae (PlasticsOne, Roanoke, VA). Surgeries were conducted under ketamine (85 mg/kg, i.p.; Bioniche Pharma, Lake Forest, IL) and xylazine (13 mg/kg, i.p.; Lloyd Laboratories, Walnut, CA) anesthesia. Guide cannulae were aimed to place the active membrane of dialysis probes (*Figure 4—figure supplement 1B*), the tip of infusion cannulae, and/or electrodes in CA1. The coordinates for microdialysis guide cannulae were: AP: -3.8, ML: ± 1.9, DV: -2.3 from the top of the skull and AP: -3.8, ML: ± 1.9, DV: -1.7 from the top of the skull for infusion guide cannulae (*Paxinos and Watson, 2004*). For EEG with intra-

hippocampal infusions, polyimide-coated stainless steel electrodes (0.15 mm OD, PlasticsOne) were attached to guide cannulae, and extended 2 mm beyond the end of guide cannulae. For EEG without intra-hippocampal infusions, the same recording electrodes were attached to guide wires with recording electrodes extending 2 mm beyond the end of the guide wire. All implants were secured with skull screws and dental cement. Following surgeries, rats were housed in groups of 2–3 for infusion and EEG experiments or singly for microdialysis experiments. All experiments were conducted 7–10 days after surgeries. The estrous phase of gonadally intact females was assessed by daily vaginal smears. Estrous phases on the day of seizure testing were Veh: 3 estrus, 1 metestrus, 1 diestrus and Let: 2 estrus, 2 diestrus. Previous studies have reported no estrous cycle-related differences in the occurrence of acute KA-induced SE (*D'Amour et al., 2015*) and we also found no differences related to the estrous cycle. All infusion sites, microdialysis probe and electrode placements were histologically verified after experiments were complete.

## KA-induced seizure testing and behavioral seizure scoring

The standard dose of KA (10 mg/kg, *Sperk et al., 1985*) was used for all experiments with EEG recording, except in gonadally intact females, in which KA was lowered to 7.5 mg/kg because 10 mg/kg in intact females resulted in very severe seizures in a pilot study. For the experiment without EEG recording, the dose was increased to 15 mg/kg to achieve the same range of seizure severity as in animals with EEG electrodes implanted. For the dialysis experiments, 1.25–5 mg/kg doses were used, as these animals experienced more severe seizures than rats without microdialysis, possibly due to tissue damage from microdialysis probes.

The testing chamber was 45 x 45 cm for microdialysis and EEG testing or 25 x 45 cm for behavior-only testing. Each rat was injected with KA (i.p., in saline) at the doses indicated above. All behavioral seizures were recorded with CCD cameras and LimeLight2 behavior tracking software (Actimetrics, Wilmette, IL). Videos were coded and seizure behaviors were later scored by experimenters blind to treatment, using criteria established by *Racine (1972)*. Because distinguishing Racine stage 1 (oral movement/chewing) from stage 2 (head nodding/waving) was often difficult in videos, these stages were grouped together as Racine 1–2 (C-HW). Staring was recorded only for the first 30 min in the study without EEG recording, because most rats in these experiments either showed no sign of seizure or progressed to more severe seizure stages after 30 min. The first >20 s period of staring was used for latency and proportion calculation. When behavioral seizure monitoring was extended to 6 hrs, the first 2 hrs were scored as above, followed by scoring of Racine 3–5 seizures to evaluate SE during the last 4 hrs. Custom Matlab scripts (https://github.com/SatoruM-Sato/seizure) were used for all seizure data processing.

## Doses of aromatase inhibitors

Fadrozole is a potent and specific non-steroidal aromatase inhibitor (*Häusler et al., 1989a*; *1989b*; *Browne et al., 1991*) that has been used previously to study how chronic aromatase inhibition influences male sex behavior (*Vagell and McGinnis, 1997*; *Huddleston et al., 2006*) and affective behaviors (*Graham and Milad, 2014*; *Carrier et al., 2015*). The doses of fadrozole used were derived from previous studies that utilized another aromatase inhibitor, vorozole, which is ~2.5 times more potent than fadrozole, and a pilot study. *Taziaux et al. (2007)* used 1 mg vorozole injected s.c. in mice to inhibit male sexual behavior. For systemic treatment, this dose is equivalent to 80 mg/kg fadrozole. We tested a lower dose (40 mg/kg) and found that it was highly effective in suppressing seizures. For intra-hippocampal infusion, we adjusted the dose for the volume of the hippocampus and estimated that 1–10 μg/side fadrozole would result in an effective concentration. This is in line with concentrations shown to inhibit aromatase activity in vitro (0.29–29 μg/ml, *Wade et al., 1994*). We then conducted a pilot study on the effects of intra-hippocampal fadrozole on KA-induced seizures using 1 and 10 μg/side, and observed significant attenuation of seizures with 10 μg/side, which was not obvious with 1 μg/side. Thus, we chose 10 μg/side for further experiments with intra-hippocampal infusion. Another non-steroidal aromatase inhibitor, letrozole, was used in experiments with gonadally intact rats. The dose of letrozole (1 mg/kg) was chosen based on a previous study of affective behavior in rats (*Kokras et al., 2014*), as well as a pharmacokinetic study of letrozole-cyclodextrin complex (*Wempe et al., 2007*).

## i.v. (tail vein) injection and intra-hippocampal infusion

For systemic administration of vehicle or an aromatase inhibitor, an i.v. catheter (24G) was inserted into the lateral tail vein, flushed with 200U heparin in 0.2 ml saline, and protected with plastic tubing before the start of EEG recording. As soon as the first hippocampal electrographic seizure subsided, the protective tubing was removed and i.v. injections were performed. For the experiment with fadrozole, 1 ml/kg saline vehicle or fadrozole HCl in saline (40 mg/ml) was slowly injected, and drug solution inside the catheter was flushed with 0.2 ml saline. Seven male rats failed to develop electrographic seizures within 45 min of KA administration. These animals were returned to their home cages and re-tested 1–2 days later. Two developed electrographic seizures on the 2nd trial and were included (1 Veh, 1 Fad), whereas 5 failed to develop seizures on 2nd trial and were excluded. Procedures were the same for the experiment with letrozole except that 19% (w/v) (2-hydroxypropyl) β-cyclodextrin in saline was used as the vehicle. In this experiment, 4 female rats that failed to develop electrographic seizures within 45 min on the 1st trial did develop seizures on the 2nd trial and thus were included (3 Veh, 1 Let). For intra-hippocampal administration of vehicle or fadrozole, rats were infused bilaterally (1 µl/side) either with vehicle (30% w/v (2-hydroxypropyl) β-cyclodextrin in saline) or fadrozole HCl (10 µg/µl), into the dorsal CA1 region at 1 µl/min through 28G internal infusion cannulae extending 2 mm beyond the implanted guide cannulae. Infusion cannulae were left in place for 1 min then replaced with an obturator.

## Intra-hippocampal EEG recording and analyses

For EEG recordings, the testing chamber was placed inside a Faraday cage. The electrode in the contralateral hippocampus served as the reference electrode. This approach, rather than using a skull screw or another epidural electrode as a reference, resulted in a cleaner EEG signal that was more resistant to motion artifacts in a pilot study. EEG electrodes were connected to a commutator mounted on a balance arm. EEG signals were filtered (high-pass: 1 Hz; low-pass: 500 Hz; notch filter at 60 Hz), amplified, and digitized at 400 Hz sampling frequency. Digitized signal was stored on a PC and later analyzed with custom Matlab scripts (https://github.com/SatoruMSato/seizure).

One day before testing, rats were acclimated to the testing chamber for 20 min. For EEG-recording with systemic (i.v.) fadrozole/letrozole or vehicle, each rat was moved to the testing chamber and baseline EEG was recorded for 20 min, then KA was injected, and recording was terminated 2 or 6 hrs after i.v. injection. For EEG recording with intra-hippocampal infusion, pre-infusion baseline EEG was recorded for 20 min, the rat was removed from the testing chamber, and drugs were infused into the hippocampus as described above. The rat was then returned to the testing chamber and EEG and time-synched video were recorded for 20 min before and 2 hrs after KA injection. To account for individual differences, we quantified EEG signals normalized to baseline for each animal. The signal from a 120 s period beginning 800 s from the start of EEG recording, shortly before either KA injection or hippocampal infusions, was used as baseline. Rats were typically quietly resting during this time after an initial period of exploration. On rare occasions when motion artifact was observed in the baseline EEG, the baseline period used for quantification was shifted by 60 s earlier or later. Signals from 50–100 Hz were not used for analysis due to the use of a 60 Hz notch filter.

Power spectrum density (PSD) was calculated by fast-Fourier transformation of EEG data for each 5 s bin. To account for differences between individuals, for example due to variation in electrode placement, we quantified seizure-related changes in power normalized to baseline for each animal (*Medvedev et al., 2000*; *Lehmkhule et al., 2009*). Quantification of seizure severity using power in a similar frequency range (20–70 Hz) has been used in multiple previous studies (e.g., *Lehmkule et al., 2009*; *Giménez-Cassina et al., 2012*; *Liu et al, 2013*; *Rook et al., 2013*) as a way of objective, efficient, and motion artifact-resistant quantification of electrographic seizures during SE. For amplitude-based seizure detection, we used 5x baseline amplitude for 30 s as criterion and PSD-based seizure detection was based on power exceeding 5x baseline for 20 s.

To evaluate time spent in seizure, we used a 5x baseline threshold, which was based on comparisons of manually detected seizures with seizures detected with various thresholds. *Figure 1—figure supplement 2* shows the similarity between manual (*Figure 1—figure supplement 2A*)-, normalized amplitude (*Figure 1—figure supplement 2B*)-, and normalized β-low γ power (*Figure 1—figure supplement 2C*)-based seizure detection. While manual and β-low γ-based detection were more sensitive to minor seizures (e.g., the 1st seizure), amplitude-based detection was more effective in

resolving seizures in the merging phase (e.g., 4000–7200 s in *Figure 1—figure supplement 2A,B*). With both amplitude- and power-based seizure detection methods, 10x baseline threshold failed to detect minor seizures (e.g., the 1st seizure), whereas 3x baseline threshold failed to resolve seizures at the onset of merging seizures (4000–5000 s). Nonetheless, the ability of fadrozole to attenuate seizure progression was evident and statistically significant regardless of the threshold used for amplitude (*Figure 1—figure supplement 2D*) or power (*Figure 1—figure supplement 2E*)-based seizure detection.

For EEG-behavior cross-correlations, normalized EEG signal (root mean square amplitude or β-low γ power) and the presence of each seizure behavior assessed from coded videos were cross-correlated for each 5 s bin with no lag. Note that differences in the strength of EEG-behavior relationships cannot be attributed solely to less frequent occurrence of certain seizure behaviors, because seizure behaviors that occur less frequently (e.g., C-HW relative to staring, or rearing relative to FLC) showed a similar strength of relationship. Thus, the higher coefficients for mild limbic relative to convulsive seizure behaviors indicate better temporal correspondence between high hippocampal electrographic activity and the occurrence of mild limbic seizures.

## Analysis of behavioral and electrographic SE

Quantitative definitions of SE vary widely in the literature (*White et al., 2010*; *D'Amour et al., 2015*; *Mishra et al., 2015*). For the purpose of this study, we identified the onset of behavioral SE when Racine 3–5 seizures (FLC, rear, fall/tonic-clonic) occurred uninterrupted for at least 30 s and continued with no more than 2 min between occurrence of Racine 3–5 seizures. Mild seizures (Racine 1–2) almost always continued after termination of an SE episode, but this was not counted as SE. If/when another 30 s period of uninterrupted Racine 3–5 seizures occurred in the same animal, SE was considered to have begun again and to last until 2 min passed with no Racine 3–5 seizures. We also evaluated electrographic SE, for which there is also no consensus definition (*Trinka et al., 2015*). We identified the onset of electrographic SE as an increase in EEG amplitude exceeding 10x baseline for at least 30 sec. After beginning, electrographic SE was considered to end if amplitude dropped below 3x baseline for 30 sec. We found that these parameters captured severe merging seizures without classifying earlier isolated seizures as SE. Also, note that while the term 'electrographic SE' has been used synonymously with non-convulsive SE (e.g., *White et al., 2010*), the parameters used here were designed to capture severe electrographic seizures that can accompany behavioral SE.

## Enzyme immunoassay for E2

We used a commercially available enzyme immunoassay (EIA) kit (Cayman Chemical, Ann Arbor, MI) for measurement of E2, with one modification: the E2 tracer was diluted by 50% with Ringer's solution (Na$^+$: 132.2, K$^+$: 4.96, Ca$^{++}$: 1.53, and Cl$^-$: 140.22 in mM) including 0.5% bovine serum albumin (BSA), which we found increased sensitivity of the assay. EIA plates were read with a Synergy 4 plate reader and a standard curve was built with a 4-parameter logistic equation using 6 concentrations of E2 standards (2.048–200 pg/ml). The intra-assay coefficient of variance was 26.0% for 5.12–200 pg/ml E2 and the inter-assay coefficient of variance was 43.1% at 13 pg/ml E2. E2 concentrations are reported directly as measured in dialysate.

## Cross-reactivity/interference tests for EIA

The EIA used in our study was tested extensively in a previous study in birds (*Remage-Healey et al., 2008*). The EIA manufacturer also tested cross-reactivity with many common steroids, and reported that the antibody used in this kit shows minimal to no cross-reactivity with non-estrogens. We conducted our own tests of cross-reactivity/interference to verify and expand on these data. As summarized in *Figure 3—figure supplement 1*, both we and the manufacturer tested testosterone, estrone, and 17α-estradiol. In addition, we tested androstenedione, other steroids known to act as neurosteroids (allopregnanolone, pregnenolone, DHEA) and glucocorticoids (corticosterone, 11-dehydrocorticosterone), all purchased from Steraloids (Newport, RI). To conduct these tests, we initially included 200 ng/ml of the steroid in 32 pg/ml of E2 standard and tested for a shift in measured E2 concentration. Assays were run in duplicates. If significant (>0.02%) cross-reactivity was suspected, we assayed a series of concentrations of the steroid to quantify cross-reactivity. All tests confirmed minimal to no cross-reactivity with compounds other than estrogens. Consistent with our

tests, the manufacturer reported <0.1% cross-reactivity for non-estrogens, and <15% for estrogens other than E2. The cross-reactivity of fadrozole was only 0.00025%, but the concentration required to inhibit aromatase was high enough to be detected, which precluded measurement of hippocampal E2 levels in fadrozole-treated rats.

## In vivo microdialysis

A microdialysis probe (1 mm active membrane, 20kDa MW cut-off) was inserted under light isoflurane anesthesia and each rat was placed in a testing chamber with food, water and bedding provided. Sterile Ringer's solution with 0.5% (w/v) BSA was perfused through the probe at 1–2 µl/min via a liquid swivel mounted on a balance arm for 3–4 hr before sample collection began. Dialysate was collected at 2 µl/min through FEP tubing at 30 min intervals, immediately frozen on dry ice and stored at -80°C until assay. Given sensitivity of the EIA used to measure E2 and a maximum flow rate of 1–2 µl/min, 30 min was the shortest interval between samples in which we could reliably measure differences in E2 concentrations. Food, water, and bedding were removed immediately before seizure induction.

## Probe recovery measurements

We quantified in vitro probe recovery using microdialysis in a bath containing Ringer's solution with 0, 0.25, or 1.0 ng/ml E2 and then comparing the E2 concentration in dialysate to the known bath concentration. Four probes were tested with ascending concentrations of E2 (i.e., 0.25 followed by 1 ng/ml) and two probes were tested with descending concentrations of E2. Immediately after the second concentration of E2, the probe was rinsed briefly in Ringer's, placed in fresh Ringer's with 0 pg/ml E2, and then two additional 30 min samples were collected to measure carryover. As little is known about the state of E2 in the brain (e.g., whether it is sequestered through binding to carrier molecules as in plasma), we did not include BSA in the bath. A pilot study using BSA in the bath resulted in much lower recovery (data not shown).

## Histology

Rats were deeply anesthetized with an overdose of Euthasol (Virbac USA, Fort Worth, TX) and transcardially perfused with 4% paraformaldehyde in 0.1 M phosphate buffer. Sections (80 µm) were cut on a vibratome, mounted on gelatin-coated slides, Nissl-stained, and examined to verify probe or infusion cannulae placement.

## Fluoro-Jade B staining

Subsets of rats used for experiments with intra-hippocampal fadrozole (n=12, 6 males and 6 females; data were combined) or systemic letrozole (n=12, all males) were sacrificed 4 d after KA seizure testing. The 4-day wait period was determined from a pilot study in which we investigated cell death at 1, 4, and 7 d after KA. Coronal sections were mounted on chrome-gelatin coated slides, air dried, and stained with Fluoro-Jade B (EMD Millipore, Temecula, CA), according to the manufacturer's instructions. Fluoro-Jade B labels dead cells (*Schmued and Hopkins, 2000*). Slides were coded and imaged by experimenters blind to treatment condition using a confocal imaging system (Zeiss 510 Meta, Thornwood, NY or Perkin Elmer UltraView LCI, Waltham, MA) with a 20x objective. For the intra-hippocampal fadrozole experiment, three consecutive sections were analyzed for each animal: the section closest to the injection site and the 2 surrounding sections. In each brain region, 3 image stacks were acquired from each subregion in both hemispheres resulting in 18 image stacks per subregion and a total of 54 stacks per rat. Image stacks were 450 µm x 450 µm x 5 µm, for a volume of $1.0125 \times 10^6$ µm$^3$ per stack. For the systemic letrozole experiment, the 2 sections immediately rostral and caudal to the EEG recording site were imaged. One (CA3, hilus) or 2 (CA1) image stacks were acquired from each subregion in both hemispheres, resulting in 4 or 8 image stacks per subregion and a total of 16 stacks per rat. Image stacks were 450 µm x 450 µm x 10 µm, for a volume of $2.025 \times 10^6$ µm$^3$ per stack. All Fluoro-Jade B-positive cells contained in each coded image stack were counted by evaluating individual optical sections using Volocity image analysis software (Improvision, Perkin Elmer). Mean densities of Fluoro-Jade B-positive cells were calculated with n as the number of animals.

## Chemicals

All chemicals were purchased from Sigma-Aldrich (St. Louis, MO) unless otherwise noted.

## Data analysis

Data are expressed as mean ± SEM, except for proportion data. N is number of animals, except in the case of microdialysis probe recovery experiments for which n = number of probes tested. EEG amplitude, PSD power and seizure duration data were analyzed with either two-way mixed (drug x time) ANOVAs with time as the repeated variable (*Figures 1* and *2*) followed by post-hoc pairwise comparisons with unpaired *t*-tests between drug groups (vehicle vs. fadrozole) at each hr of testing with Bonferroni correction, or with unpaired *t*-tests between overall means for drug groups (*Figure 5*). Cross-correlations were compared between seizure behaviors with one-way ANOVAs, followed by Newman-Keuls pairwise comparisons between each behavior and unpaired *t*-tests between drug groups (vehicle vs. fadrozole) with Bonferoni correction (*Figure 2* and *Figure 5*). Endogenous aromatase activity was assessed by comparing E2 concentration before vs. after 4-dione retrodialysis with a paired *t*-test (*Figure 3*). Effects of KA-induced seizures on hippocampal E2 were analyzed with one-way repeated measures ANOVAs with time (hr) as the repeated variable followed by pairwise comparisons with post-hoc paired *t*-tests with Bonfferoni correction (*Figure 4*). For behavioral seizure analysis (*Figure 6A*), proportion data for rats reaching each stage of seizure behavior were compared with $\chi^2$ tests. The remaining group comparisons were analyzed with unpaired *t*-tests (e.g., behavioral seizure latency and duration, analyses of $1^{st}$ seizures, Fluoro-Jade B cell counts). Because male and female rats received different doses of KA in the systemic letrozole experiment (*Figures 7*, *8*), KA dose was included as a covariate and results were analyzed by ANCOVA to allow for concurrent analysis of data from both sexes. In all cases, p<0.05 was considered statistically significant. To use the minimum number of animals necessary, we analyzed data once n=4~5 (or ~8 for the non-EEG/dialysis experiment) was reached, and the sample size was increased if a statistical trend (p<0.2) was observed. Final effect sizes were generally large (Cohen's d≈0.8) and never d<0.69 (*Supplementary file 1*).

## Acknowledgments

Research was supported by National Institute of Neurological Disorders and Stroke grant NS037324 and the Office of Research on Women's Health, the Northwestern University High Throughput Analysis Laboratory and Biological Imaging Facility. The authors are grateful to Renee M May, Andrew F McCollum, Monica J Rondinelli, Madeline M Roberson, Kavish Gupta, Merav Stein, Matthew J Fligiel, Marina Murashev, Nicholas F Hug, Jenny (Yucong) Zhang, Emma G Zblewski and Kerry M McFadden for assistance with behavioral and electrographic seizure analyses and histology.

## Additional information

### Funding

| Funder | Grant reference number | Author |
| --- | --- | --- |
| National Institute of Neurological Disorders and Stroke | NS037324 | Catherine S Woolley |
| Office of Research on Women's Health | NS037324 | Catherine S Woolley |

The funders had no role in study design, data collection and interpretation, or the decision to submit the work for publication.

### Author contributions

SMS, Designed experiments, Conducted experiments and analyzed the data, Prepared the manuscript, Drafting or revising the article; CSW, Designed experiments, Prepared the manuscript, Analysis and interpretation of data, Drafting or revising the article

**Author ORCIDs**

Catherine S Woolley, http://orcid.org/0000-0002-8069-2646

**Ethics**

Animal experimentation: All animal procedures were performed in accordance with the National Institutes of Health Guide for the Care and Use of Laboratory Animals and were approved by the Northwestern University Animal Care and Use Committee. Animal Study Protocol IS00000520 (expires 6/26/2017) Animal Welfare Assurance A3283-01

## Additional files

**Supplementary files**

• Supplementary file 1. Summary of effect sizes. Each table lists significant effects shown in each Figure with the respective mean sample size and Cohen's d statistic for each effect.

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
