## [Decision Letter]

Thank you for submitting your work entitled "Acute Inhibition of Neurosteroid Estrogen Synthesis Suppresses Status Epilepticus in an Animal Model" for consideration by *eLife*. Your article has been reviewed by two peer reviewers, and the evaluation has been overseen by Gary Westbrook as the Senior Editor. The reviewers have discussed the reviews with one another and the Editor has drafted this decision to help you prepare a revised submission.

Summary:

This study examines the role of an aromatase inhibitor (fadrozole) in the response of gonadectomized male and female rats to the convulsant kainic acid. The results suggest that the inhibitor decreased the severity of the seizures, and the authors suggest that the reason is that hippocampal estradiol levels were reduced. The authors measure estradiol by microdialysis and report that estradiol increased after kainic acid injection. The authors conclude that estradiol synthesis in hippocampus increases after kainic acid, and this increases the severity of seizures. They also suggest that these data provide a foundation for a new therapeutic approach to SE. The paper has many strengths: it is well written and the methods are clearly laid out. References are thorough. The results generally show the effects of fadrozole to decrease the effects of kainic acid. It is also interesting that estradiol levels in hippocampus appear to rise as seizures occur. There are concerns however, most importantly that the experiments were done in gonadectomized animals thus the relevance to intact animals is unclear. We think this point requires additional experiments (see point 2 below), which we hope would be feasible within the *eLife* policy of 2 months.

1) Specificity of the drug. Regarding fadrozole, aromatase inhibitors do not only affect estradiol synthesis. The authors do acknowledge this in the Discussion, but it seems hard to dismiss.

2) The exclusive use of gonadectomized animals. The use of gonadectomized rats undermines the broad relevance of the study, and also the therapeutic relevance. Gonadectomy leads to numerous changes in the brain and periphery, and hippocampal circuits are likely to dramatically change, as the authors are well aware. This could be dependent on the time after gonadectomy also, as the authors are also aware (what was this time?). Seizure susceptibility in the female and male should dramatically rise after gonadectomy, in light of work showing that the metabolites of progesterone and testosterone enhance GABAergic inhibition and they are largely removed by gonadectomy. Although intact rats are difficult to work with but this point is very important when the suggestion is made that the work could lead to a therapeutic strategy in humans with SE. Note that it is also true that in most patients it won't be possible to treat before SE begins. Therefore, the therapeutic relevance is not too clear at this point. Overall, one wonders how much the results would be generalizeable to intact animals. Experiments in intact animals are necessary for obvious reasons: most patients won't be gonadectomized.

In addition, the microdialysis results suggest that the estradiol involved in the seizure-promoting effect are within the physiological range, a very important line of argument, but as the authors point out, microdialysis is not an exact measurement of concentrations at synapses.

3) Seizure and status epilepticus analysis. The analyzed appear to be the first two hours after kainic acid injection, when there is a period of individual nonconvulsive and convulsive seizures and then onset of SE. The authors don't separate the initial individual seizures and SE, pooling everything. However, if the authors want to study SE, the seizures leading up to SE don't seem important. In SE is studied, the latency to onset, the duration, and incidence should be measured. It seems that animals with fadrozole almost always failed to exhibit SE (from Figure 6—figure supplement 1, the heat maps), supporting the authors' conclusions, so it would be good to do an analysis of SE incidence at least.

Because the authors pool all data, and only study the first two hours of SE it is hard to address power because it may have surged after the 2 hours, when SE becomes most severe (at least by some reports). When a seizure begins and ends after KA is very hard to say because one transitions to another. For example, seizures after KA often transition from stage 1 to a stage 2 or stage 3 to 4 – is this two seizures or one? Where does each start? These issues need better definition and discussion.

4) EEG analyses. If sites of electrode tips differed, which is the case here, then the EEG will be substantially different because the different layers of the hippocampus have different spectral characteristics. For example, γ is best detected in cell layers and θ is best detected in layers such as stratum lacunosum-moleculare. Therefore, it is not clear one can compare the EEG across animals, as the authors have. Because gliosis at the electrode site (and other aspects of the recordings) differs from animal to animal, it is hard to compare EEG amplitude across animals.

It's interesting to look at the relationship between EEG and seizure-associated behavior, but the fourth paragraph of the subsection “Extra-gonadal aromatase inhibition acutely suppresses seizures” does not lead to a strong conclusion because it is not clear one can say anything about seizure spread based on the cross correlation. Spread is something one needs look at with multiple electrodes, determining where seizures begin – i.e. one electrode starts before another.

In addition, EEG power was used as a surrogate for seizures. Computer-measured EEG power has low sensitivity and specificity for seizure detection. Muscle activity and motor movements contribute to EEG power, and seizures cause muscle activity and movement. Also hippocampal seizures (in humans, rats or mice) are rhythmic spike wave discharges in θ range, thus it is unclear why the authors focused on β power.

5) A key goal of treatments for status epilepticus is to shorten its duration. Neuronal injury and death correlate with seizure duration. However, the authors did not demonstrate a reduction in seizure (status epilepticus) duration. Interestingly cell death was not prevented by this treatment either. This point requires additional documentation or discussion.

---

## [Author Response]

Summary:

*This study examines the role of an aromatase inhibitor (fadrozole) in the response of gonadectomized male and female rats to the convulsant kainic acid. The results suggest that the inhibitor decreased the severity of the seizures, and the authors suggest that the reason is that hippocampal estradiol levels were reduced. The authors measure estradiol by microdialysis and report that estradiol increased after kainic acid injection. The authors conclude that estradiol synthesis in hippocampus increases after kainic acid, and this increases the severity of seizures. They also suggest that these data provide a foundation for a new therapeutic approach to SE. The paper has many strengths: it is well written and the methods are clearly laid out. References are thorough. The results generally show the effects of fadrozole to decrease the effects of kainic acid. It is also interesting that estradiol levels in hippocampus appear to rise as seizures occur. There are concerns however, most importantly that the experiments were done in gonadectomized animals thus the relevance to intact animals is unclear. We think this point requires additional experiments (see point 2 below), which we hope would be feasible within the eLife policy of 2 months.*

1) Specificity of the drug. Regarding fadrozole, aromatase inhibitors do not only affect estradiol synthesis. The authors do acknowledge this in the Discussion, but it seems hard to dismiss.

To address this issue, we performed additional experiments with a second aromatase inhibitor, letrozole, and found that letrozole also suppressed seizures as fadrozole did. These new results, together with the seizure-induced increase in hippocampal E2 observed with microdialysis and the well-known effects of E2 to acutely promote neural activity in the hippocampus, make an estrogen-based explanation of the seizure-suppressing effects of aromatase inhibitors the most parsimonious one.

That said, it is possible that inhibiting estrogen synthesis in the brain has the indirect consequence of altering levels of other neurosteroids, particularly estrogen precursors, and we cannot rule out the possibility that such indirect effects may also influence seizures. We acknowledge this in the Discussion (subsection “Mechanisms by which estrogens acutely modulate hippocampal activity”, first paragraph) and have added a statement that future studies will be necessary to address this possibility.

2) The exclusive use of gonadectomized animals. The use of gonadectomized rats undermines the broad relevance of the study, and also the therapeutic relevance. Gonadectomy leads to numerous changes in the brain and periphery, and hippocampal circuits are likely to dramatically change, as the authors are well aware. This could be dependent on the time after gonadectomy also, as the authors are also aware (what was this time?). Seizure susceptibility in the female and male should dramatically rise after gonadectomy, in light of work showing that the metabolites of progesterone and testosterone enhance GABAergic inhibition and they are largely removed by gonadectomy. Although intact rats are difficult to work with but this point is very important when the suggestion is made that the work could lead to a therapeutic strategy in humans with SE. Note that it is also true that in most patients it won't be possible to treat before SE begins. Therefore, the therapeutic relevance is not too clear at this point. Overall, one wonders how much the results would be generalizeable to intact animals. Experiments in intact animals are necessary for obvious reasons: most patients won't be gonadectomized.

In the previous submission, we had used gonadectomized animals to eliminate circulating gonadal hormones and focus on E2 synthesized in the brain. However, we concur that the therapeutic potential of our results would be increased by demonstrating that aromatase inhibitors also suppress seizures in gonadally intact animals. To address this, we performed new experiments in which we used systemic administration of an aromatase inhibitor that is used clinically, letrozole, in gonadally intact animals (males and females) after seizures had begun. We also extended the period of seizure monitoring in this experiment to 6 hours, to address other concerns raised below regarding analysis of SE. In brief, these new studies showed that acute systemic aromatase inhibition effectively suppressed seizures in gonadally intact animals, similar to acute systemic aromatase inhibition in gonadectomized animals. These findings are described in a new section of the Results (”Systemic aromatase inhibition acutely suppresses seizures in gonadally intact animals”) and are shown in new Figure 7 and 8.

One difference we observed between gonadectomized and gonadally intact animals is that behavioral seizures were much *less* severe in gonadectomized animals of both sexes (electrographic seizures did not appear to differ, however). Owing to the Reviewers’ comment, we mention this in the revised Results (in the aforementioned section) and include a new supplemental figure (Figure 8—figure supplement 1) to illustrate the difference. Thus, while it has been shown that metabolites of progesterone and testosterone are anticonvulsant in several seizure models, the physiological source of these steroids that suppress seizures in vivo may not be the gonads.

To answer the reviewers’ question about the duration of gonadectomy, animals were used for experiments 7-10 days after surgery (Materials and methods, first paragraph).

In addition, the microdialysis results suggest that the estradiol involved in the seizure-promoting effect are within the physiological range, a very important line of argument, but as the authors point out, microdialysis is not an exact measurement of concentrations at synapses.

To further clarify this important issue, we have revised the relevant section of the Discussion (subsection “Mechanisms by which estrogens acutely modulate hippocampal activity”, last paragraph) to explain how our measurements of E2 from microdialysis relate to the concentrations of E2 that have been shown to acutely promote neural activity when bath-applied to hippocampal slices.

3) Seizure and status epilepticus analysis. The analyzed appear to be the first two hours after kainic acid injection, when there is a period of individual nonconvulsive and convulsive seizures and then onset of SE. The authors don't separate the initial individual seizures and SE, pooling everything. However, if the authors want to study SE, the seizures leading up to SE don't seem important. In SE is studied, the latency to onset, the duration, and incidence should be measured. It seems that animals with fadrozole almost always failed to exhibit SE (from Figure 6—figure supplement 1, the heat maps), supporting the authors' conclusions, so it would be good to do an analysis of SE incidence at least.

This is an important point and we thank the Reviewers for suggesting additional analyses focused specifically on SE. To address this, we quantified SE incidence, latency to onset, and time spent in SE both for the 2 hr dataset previously obtained with systemic fadrozole in gonadectomized animals and the new 6 hr dataset obtained with systemic letrozole in gonadally intact animals. We evaluated SE in two ways: based on sustained periods of Racine 3-5 convulsive seizures (behavioral SE) and based on sustained periods during which EEG amplitude exceeded 10x baseline (electrographic SE). In brief, both types of analysis showed that aromatase inhibition suppressed SE in both experiments. The results with fadrozole in gonadectomized animals are described in a new section of the Results (subsection”Extra-gonadal aromatase inhibition acutely suppresses seizures”, last paragraph) and shown in new Figure panels 2H, J and the results with letrozole in gonadally intact animals are described in a new section of the Results (subsection “Systemic aromatase inhibition acutely suppresses seizures in gonadally intact animals”, third paragraph) and a new Figure 8. The Materials and methods have been revised to include a description of SE analyses (subsection “Analysis of behavioral and electrographic SE”).

Because the authors pool all data, and only study the first two hours of SE it is hard to address power because it may have surged after the 2 hours, when SE becomes most severe (at least by some reports).

This was a helpful comment, and prompted us to extend seizure monitoring from 2 to 6 hrs in additional experiments with letrozole in gonadally intact animals. As evident in the EEG recordings, normalized power spectra, and summary data shown in Figure 7, seizure activity reached its peak around 1.5 hr after the first electrographic seizure was detected in the hippocampus (~2 hrs after KA injection) and then gradually declined. Neither EEG amplitude nor power increased further after 2 hrs. Also, as these figures show, the effect of aromatase inhibition to suppress seizure activity extended for the duration of 6 hrs of recording and thus aromatase inhibitors did not simply delay seizure escalation.

When a seizure begins and ends after KA is very hard to say because one transitions to another. For example, seizures after KA often transition from stage 1 to a stage 2 or stage 3 to 4 – is this two seizures or one? Where does each start? These issues need better definition and discussion.

We concur that it can be difficult to identify isolated bouts of seizure, especially as seizures progress to become more severe. For behavioral seizures, we did not attempt to identify individual seizure bouts, but instead scored specific seizure behaviors (Racine stages) per second of each video and report% time in seizure in 5 min increments (e.g., Figure 2). For the heatmaps in Figure 6—figure supplement 1, we show the highest stage of seizure behavior reached per minute, as indicated in the Results text (subsection “Inhibition of hippocampal E2 synthesis suppresses behavioral seizures”, first paragraph) and related figure legend. For identification of electrographic seizures, we used a threshold approach based on normalized EEG amplitude or power, as illustrated in Figure 1—figure supplement 2. We report normalized EEG amplitude, power, or% time in seizure in 5 min (Figure 1) or 10 min (Figure 7) bins throughout each recording period (2 or 6 hrs). Thus we did not attempt to identify the beginning and end of an electrographic seizure. The one exception to this is our approach to identifying periods of SE (which did have an onset and an end, necessitated by the suggestion above to quantify SE latency to onset and duration), as detailed in the revised Materials and methods (subsection “Analysis of behavioral and electrographic SE”).

4) EEG analyses. If sites of electrode tips differed, which is the case here, then the EEG will be substantially different because the different layers of the hippocampus have different spectral characteristics. For example, γ is best detected in cell layers and θ is best detected in layers such as stratum lacunosum-moleculare. Therefore, it is not clear one can compare the EEG across animals, as the authors have. Because gliosis at the electrode site (and other aspects of the recordings) differs from animal to animal, it is hard to compare EEG amplitude across animals.

This concern raised by the reviewers, individual differences in electrode placement, etc., is the reason we quantified EEG amplitude and power relative to baseline for each individual animal. Our approach was based on that in Medvedev et al., 2000, Brain Res Bull; Lehmkhule et al., 2009, J Neurophysiol and also used by others (e.g., Gimenez-Cassina et al., 2012, Neuron; Liu et al., 2013, Neuron; Rook et al., 2013, Biol Psychiatry). We have revised the Materials and methods section (subsection “Intra-hippocampal EEG recording and analyses”, second and third paragraphs) to further clarify this and to emphasize that EEG amplitude and power results are reported as changes relative to baseline (i.e., normalized amplitude or normalized power). In addition, we have made revisions throughout the Results text, figures, and figure legends to be clearer that we report seizure-related changes in EEG amplitude or power, which is what we compare across animals. We have not compared raw EEG amplitude or power across animals.

It's interesting to look at the relationship between EEG and seizure-associated behavior, but the fourth paragraph of the subsection “Extra-gonadal aromatase inhibition acutely suppresses seizures” does not lead to a strong conclusion because it is not clear one can say anything about seizure spread based on the cross correlation. Spread is something one needs look at with multiple electrodes, determining where seizures begin – i.e. one electrode starts before another.

We removed reference to seizure spread from this sentence and eliminated the section of the Discussion that had addressed this idea.

In addition, EEG power was used as a surrogate for seizures. Computer-measured EEG power has low sensitivity and specificity for seizure detection. Muscle activity and motor movements contribute to EEG power, and seizures cause muscle activity and movement. Also hippocampal seizures (in humans, rats or mice) are rhythmic spike wave discharges in θ range, thus it is unclear why the authors focused on β power.

Our use of β-low γ power as an approach to detect seizures was driven partly by the existing literature and partly by our observations. In the literature, a KA-induced increase in power in the β-low γ range is well-documented both in vivo and in vitro (Fisahn et al., 2004, J Neurosci; Medvedev et al., 2000, Brain Res Bull; Lévesque et al., 2009, Neurobiol Dis) and quantification of seizure severity using power in a similar frequency range (20-70 Hz) has been used by others (e.g., Lehmkhule et al., 2009, J Neurophysiol; Gimenez-Cassina et al., 2012, Neuron; Liu et al., 2013, Neuron; Rook et al., 2013, Biol Psychiatry) as a way of objective, efficient, and motion artifact-resistant quantification of electrographic seizures during SE. This reasoning is noted in the Results (subsection “Extra-gonadal aromatase inhibition acutely suppresses seizures”, second paragraph) as well as Materials and methods (subsection “Intra-hippocampal EEG recording and analyses”, third paragraph). In addition, our pilot studies showed that β-low γ power was relatively resistant to motion artifacts compared to lower frequency ranges, while being more broadly sensitive to seizures than higher frequency ranges. The utility of changes in β-low γ power for detecting seizures is illustrated in Figure 1—figure supplement 1 and Figure 1—figure supplement 2. Note also that we quantified and reported seizure-related increases in power in 3 frequency ranges (δ-θ, β-low γ, ripple) and report all results in the text.

5) A key goal of treatments for status epilepticus is to shorten its duration. Neuronal injury and death correlate with seizure duration. However, the authors did not demonstrate a reduction in seizure (status epilepticus) duration. Interestingly cell death was not prevented by this treatment either. This point requires additional documentation or discussion.

Our primary motivation for evaluating Fluoro-Jade B positive cells was to determine whether aromatase inhibition during seizures might produce a detrimental outcome by eliminating estrogen neuroprotection (Hoffman et al., 2003, Exp Neurol; Schauwecker et al., 2009, Brain Res). However, we found no statistically significant effects of aromatase inhibition, despite the fact that average numbers of Fluoro-Jade B positive cells were slightly lower in aromatase inhibitor-treated animals (consistent with their milder seizures). One potential explanation for this is that milder seizures following aromatase inhibition, which would be expected to result in less cell death as the Reviewers point out, is balanced by loss of estrogen neuroprotection, producing roughly equivalent seizure-related cell death. We have added this suggestion to the Discussion and noted that future, more detailed studies will be required to fully address it (subsection “Therapeutic potential of acute aromatase inhibitor treatment”).